# Promoter-proximal RNA polymerase II termination regulates transcription during human cell type transition

Kseniia Lysakovskaia[1,2], Arjun Devadas[1,2], Björn Schwalb[1], Michael Lidschreiber ®[1] ✉ & Patrick Cramer ®[1] ✉

Metazoan gene transcription by RNA polymerase II (Pol II) is regulated in the promoter-proximal region. Pol II can undergo termination in the promoter-proximal region but whether this can contribute to transcription regulation in cells remains unclear. Here we extend our previous multiomics analysis to quantify changes in transcription kinetics during a human cell type transition event. We observe that upregulation of transcription involves an increase in initiation frequency and, at a set of genes, a decrease in promoter-proximal termination. In turn, downregulation of transcription involves a decrease in initiation frequency and an increase in promoter-proximal termination. Thus, promoter-proximal termination of Pol II contributes to the regulation of human gene transcription.

Transcription by RNA polymerase II (Pol II) is an essential process for establishing cellular identity and function and is tightly regulated during initiation and early elongation[1]. In metazoan cells, Pol II frequently pauses in the region proximal to the promoter, 30–60 bp downstream of the transcription start site (TSS)[2,3]. Promoter-proximally paused Pol II is stabilized through its association with two protein complexes, DRB sensitivity-inducing factor and negative elongation factor[4–7]. The kinase activity of the positive transcription elongation factor b (P-TEFb) is required for the release of paused Pol II into productive elongation and for full-length transcript synthesis[8–15]. There is also evidence that Pol II transcription can terminate in the promoter-proximal region[16–27] but whether such premature termination is frequently used in cells to regulate gene transcription remains unclear.

Promoter-proximal Pol II pausing has been extensively studied using the relative ratio of Pol II occupancy in the promoter-proximal region to the gene body, described as the pausing index or traveling ratio[28–36]. On the basis of such analysis, it has been suggested that promoter-proximal Pol II pausing is regulated during various cellular signaling pathways, including environmental stress, the immune response and developmental and differentiation signals, ensuring a rapid and coordinated transcriptional response[37]. A limitation of using only Pol II occupancy measurements for studying transcription

regulation is that the density of Pol II on genes depends not only on the number of polymerases that initiate transcription per unit time but also on their elongation velocity[38]. Therefore, Pol II occupancy cannot provide kinetic insights and does not allow for conclusions about transcription regulation, which is because of changes in Pol II kinetics.

To overcome this limitation, we previously developed a multiomics approach that combines Pol II occupancy profiling with measurements of new RNA synthesis, allowing us to derive the productive initiation frequency and the apparent pause duration of Pol II for actively transcribed genes[39,40]. We use the terms 'productive' and 'apparent' because an unknown fraction of polymerases might terminate in the promoter-proximal region[16,22,23,25–27]. This approach, however, assumed that promoter-proximal Pol II termination occurs rarely, as suggested by prior findings showing stable Pol II pausing in the promoter-proximal region[33,36,41–43]. In case the promoter-proximal termination fraction is larger, our original model is inadequate to describe transcription kinetics. Indeed, more recent studies reported rapid turnover of promoter-proximal Pol II, indicating dynamic cycles of transcription initiation and promoter-proximal termination that occur independent of the transcriptional activity or the pausing status of a gene[44,45]. Moreover, a short residence time of paused Pol II and an increase in promoter-proximal termination were indicated

[1]Department of Molecular Biology, Max Planck Institute for Multidisciplinary Sciences, Göttingen, Germany. [2]These authors contributed equally: Kseniia Lysakovskaia, Arjun Devadas. ✉e-mail: michael.lidschreiber@mpinat.mpg.de; patrick.cramer@mpinat.mpg.de

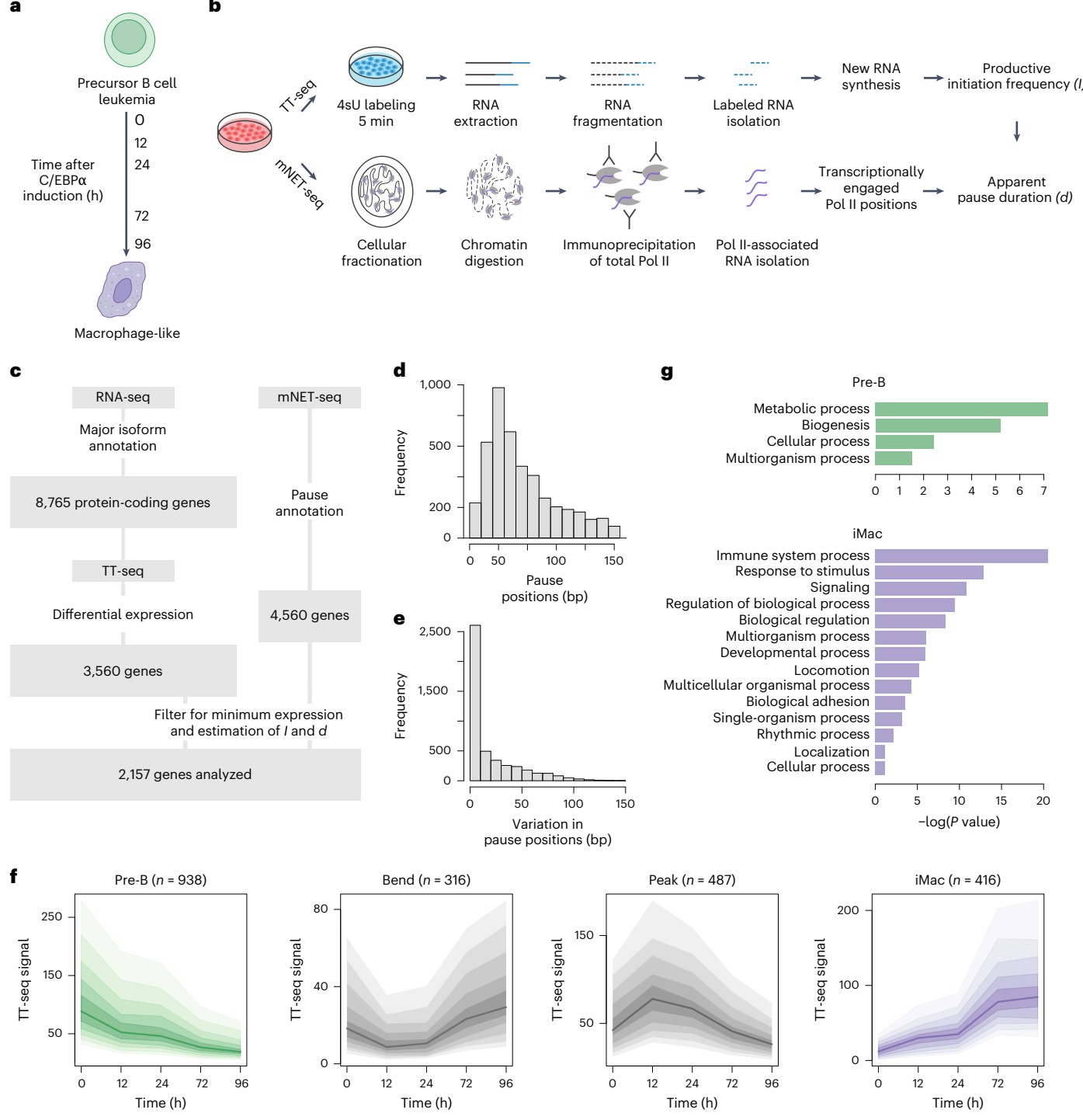

**Fig. 1 | Multiomics analysis of human cell type transition. a**, Overview of the human cell transdifferentiation system. Precursor leukemia B cells undergo transdifferentiation into macrophage-like cells upon estrogen-inducible C/EBPα overexpression[48]. Time points of cell collection after induction are shown. **b**, Schematic representation of the multiomics approach[39,40] used to infer productive initiation frequency (*I*) and apparent pause duration (*d*) from TT-seq and mNET-seq data (Methods). **c**, Schematic representation of transcript annotation. **d**, Histogram showing the distribution of Pol II promoter-proximal pause positions. Data are shown for 4,560 annotated protein-coding genes with determined pause position at at least one time point during transdifferentiation based on mNET-seq data (Methods). **e**, Histogram showing the variation of the

Pol II promoter-proximal pause positions determined at different time points of transdifferentiation. Data are shown for 4,560 protein-coding genes as in **d**. **f**, Representation of the clustering of 2,157 selected differentially expressed protein-coding genes based on the TT-seq data. Changes in RNA synthesis during transdifferentiation occurred in four defined patterns, termed pre-B (downregulated), bending, peaking and iMac (upregulated). Dark lines in the middle show the median TT-seq signal for each gene group. The outermost borders of the shaded area show the 25th and 75th percentiles. The borders in between represent 5% increments in the percentiles. **g**, GO analysis results for pre-B and iMac genes.

to occur during gene downregulation upon hyperosmotic stress in human cells[46,47]. However, it remains to be fully elucidated whether promoter-proximal Pol II termination is a regulated process that can control gene transcription in cells.

In this study, we used an established transdifferentiation system[48–50] to quantify changes in kinetic parameters underlying Pol II transcription regulation during a human cell type transition. We first estimated transcription kinetic parameters, including the productive initiation frequency and the apparent pause duration at different time points of transdifferentiation with the use of transient transcriptome sequencing (TT-seq) and mammalian nascent elongating transcript sequencing (mNET-seq) data. We then measured the half-life of promoter-proximal Pol II with the use of chromatin immunoprecipitation (ChIP)-nexus experiments following inhibition of transcription initiation. This extended multiomics approach allowed us to estimate the fraction of Pol II that undergoes promoter-proximal termination for each gene. Our results establish promoter-proximal Pol II termination as a regulatory mechanism that contributes to the regulation of genes in human cells.

## Results

### Multiomics analysis of human cell type transition

To investigate the strategies cells use to regulate Pol II transcription, we used a previously reported transdifferentiation system[48–50]. With this system, human precursor leukemia B cells (BLaER1) are converted into macrophage-like cells by estrogen-inducible CCAAT/enhancer binding protein α (C/EBPα) overexpression within 96–168 h (ref. [48]). We selected time points of 0, 12, 24, 72 and 96 h, which showed the most pronounced changes in gene expression after transdifferentiation induction on the basis of publicly available RNA sequencing (RNA-seq) data[50] (Fig. 1a). To confirm the efficiency of transdifferentiation, we monitored the expression of B cell-specific and macrophage-specific markers using quantitative PCR with reverse transcription (RT–qPCR) and obtained 70–90% of transdifferentiated cells at 96 h (Extended Data Fig. 1a), in agreement with previously performed gene expression and fluorescence-activated cell sorting analyses[48,50].

To estimate Pol II kinetic parameters during transdifferentiation, we used our previously established multiomics approach, which combines TT-seq and mNET-seq data[39,40]. TT-seq uses metabolic RNA labeling to provide an unbiased genome-wide view of RNA synthesis[51]. TT-seq allows estimation of the productive initiation frequency ($I$) (Methods and Fig. 1b), which is defined as the number of Pol II enzymes that initiate transcription, successfully pass the promoter-proximal region and enter productive elongation[39,40]. mNET-seq provides genome-wide occupancy of Pol II associated with the nascent transcript and can be used to measure the amount of Pol II located in the promoter-proximal region[52,53]. The ratio of mNET-seq to TT-seq signal allows estimation of the apparent pause duration ($d$)[39,40] (Methods and Fig. 1b), which reflects the total time that the promoter-proximal region is occupied by Pol II between two initiation events that successfully lead to productive elongation. This does not necessarily mean that one polymerase pauses for the entire estimated time but could also mean that a subpopulation of polymerases terminates early in the promoter-proximal region[40].

We conducted mNET-seq experiments with an antibody recognizing total Pol II (Methods) at 0, 12, 24, 72 and 96 h after transdifferentiation induction for two independent biological replicates (Pearson correlation coefficient = 0.96–0.99) (Extended Data Fig. 1b). We used published TT-seq data[50] for the same time points of transdifferentiation. TT-seq and mNET-seq data showed near complete downregulation of B cell-specific and upregulation of macrophage-specific gene transcription, confirming our RT–qPCR data (Extended Data Fig. 1a,c).

### Definition of promoter-proximal regulatory regions

To determine unambiguous promoter-proximal pause positions of Pol II, we created an annotation containing only the major transcribed isoform of each protein-coding gene using RNA-seq data, resulting

in a total of 8,765 genes (Methods and Fig. 1c). We extracted the positions of promoter-proximally paused polymerases on the basis of the maximum mNET-seq signal within 250 bp downstream of the TSS and retained only pause sites that were above five times the median signal in the same window (Methods). Using this approach, we were able to determine the pause position for a total of 4,560 annotated genes at at least one time point during transdifferentiation (Fig. 1c and Extended Data Fig. 1d).

The median pause position was found to be 72 bp (mode 44 bp) downstream of the TSS (Fig. 1d), consistent with previous findings in human cells[40,54,55]. Furthermore, for the majority of annotated genes, the derived pause positions varied by less than 10 bp over the time course of transdifferentiation (Fig. 1e). Taken together, this analysis showed that the transdifferentiation system used here is well suited to study the regulation of Pol II transcription during human cell type transition, mainly because of its simplicity and high efficiency. In addition, the system is suitable for the analysis of the transcription regulatory events that occur in the promoter-proximal region, particularly because we observed a strong promoter-proximal Pol II signal for a substantial number of genes.

### RNA synthesis changes during transdifferentiation

To examine changes in RNA synthesis during transdifferentiation, we performed differential expression analysis on the basis of TT-seq data (|fold change (FC)| > 1.5, adjusted $P < 0.05$) (Methods and Extended Data Fig. 1e). We identified 3,560 differentially expressed (DE) genes, of which 2,157 had a determined pause position and showed sufficient TT-seq and mNET-seq signal at a dominant isoform to allow estimation of the kinetic parameters $I$ and $d$ (Fig. 1c). During transdifferentiation, RNA synthesis of these selected DE genes changed in four distinct directions: downregulated ($n = 938$), bending ($n = 316$), peaking ($n = 487$) and upregulated ($n = 416$) (Fig. 1f).

We named downregulated genes as pre-B and upregulated genes as iMac, according to the transdifferentiation stages at which they showed maximum RNA synthesis (Fig. 1f). Gene Ontology (GO) analysis revealed that pre-B genes were enriched for biogenesis, metabolic and cellular processes, the hallmarks of actively transcribing cancer cells, whereas iMac genes were enriched for immune and macrophage-specific functions (Fig. 1g), validating our gene classification. Thus, we identified two biologically relevant groups of genes that significantly change their RNA synthesis during transdifferentiation in different ways and are suitable for investigating promoter-proximal regulatory mechanisms underlying upregulation and downregulation of transcription during cell type transition.

### Pre-B genes show promoter-proximal transcription regulation

We first examined the transcription kinetics of the pre-B genes ($n = 938$; Fig. 1f), which were downregulated during transdifferentiation. Metagene analysis revealed a decrease in TT-seq signal throughout the gene from 0 to 96 h, whereas the mNET-seq signal was mainly diminished in the gene body compared to the promoter-proximal region (Fig. 2a,b). Accordingly, $I$ decreased significantly from 0 to 96 h of transdifferentiation (median $FC_{0–96} = 0.25$, $P < 2.2 \times 10^{-16}$; Fig. 2c, left), whereas the promoter-proximal occupancy of Pol II quantified from the mNET-seq signal remained largely unchanged (Fig. 2c, middle). Furthermore, we observed a significant increase in $d$ from 0 to 96 h (median $FC_{0–96} = 4.46$, $P < 2.2 \times 10^{-16}$), suggesting that transcriptionally engaged Pol II is retained in the promoter-proximal region without proceeding into productive elongation (Fig. 2c, right).

We, therefore, assessed the genome-wide occupancy of P-TEFb, which releases paused Pol II into productive elongation[8–15]. We carried out ChIP-seq experiments for P-TEFb subunits cyclin-dependent kinase 9 (CDK9) and cyclin T1 for two independent biological replicates (Pearson correlation coefficient = 0.98–0.99) at 0, 24 and 96 h after transdifferentiation induction (Extended Data Fig. 2a,b).

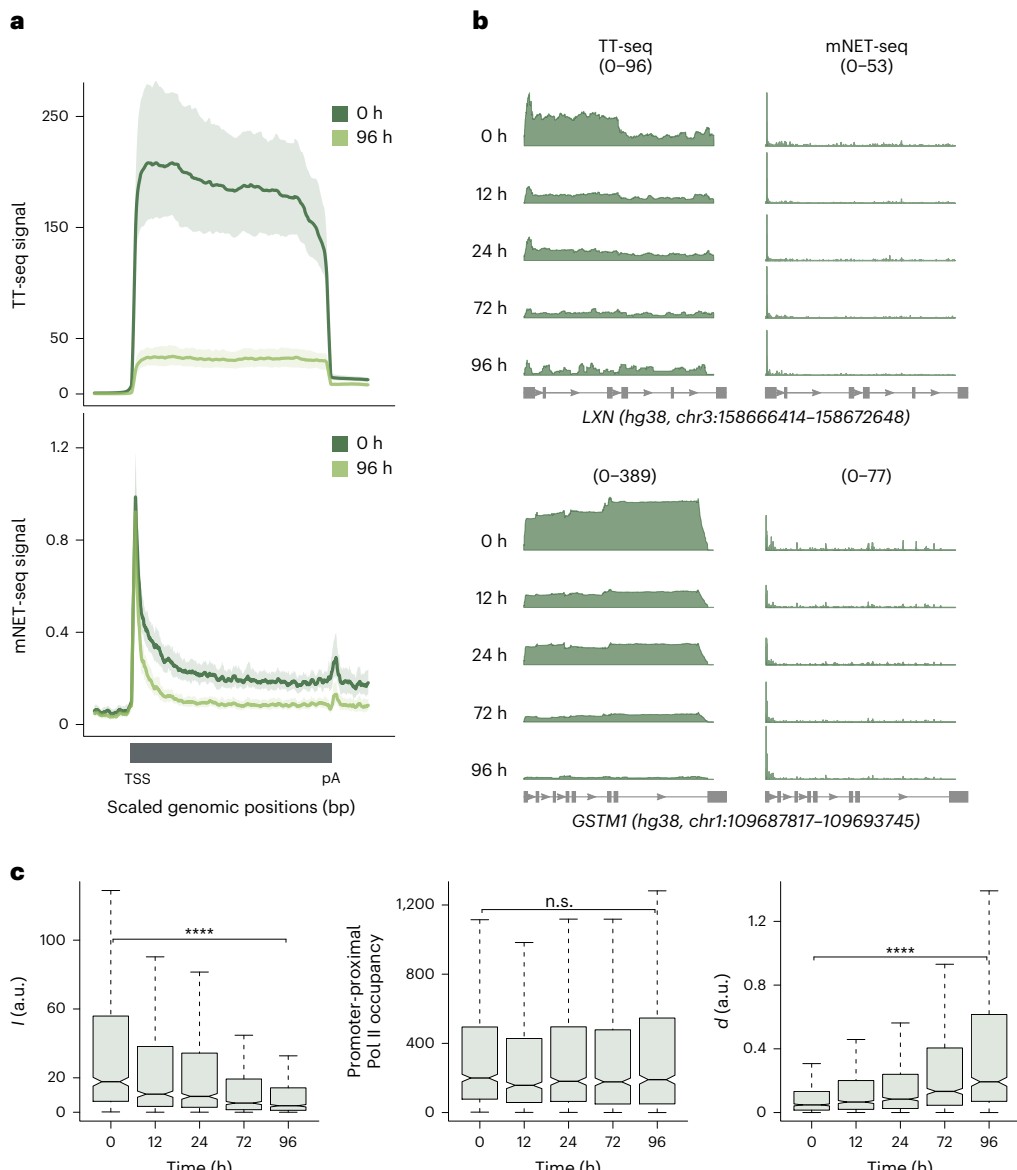

**Fig. 2 | Pre-B genes show promoter-proximal transcription regulation.**
**a**, Metagene profiles comparing TT-seq and mNET-seq signals between 0 h (dark green) and 96 h (light green) after transdifferentiation induction for pre-B genes. Both TT-seq and mNET-seq signals were averaged for 938 pre-B genes and scaled between the TSS and poly(A)-site. Data from two biological replicates are merged for illustration. Solid lines represent the averaged signal and shaded regions show the 95% confidence interval of the mean. **b**, Representative examples of pre-B genes (*LXN* and *GSTM1*). TT-seq and mNET-seq data collected at 0, 12, 24, 72 and 96 h after transdifferentiation induction are shown. Data from two biological replicates are merged for illustration. **c**, Box plots showing estimates of kinetic parameters $I$ (left) and $d$ (right) and promoter-proximal Pol II occupancy (middle) for pre-B genes ($n = 938$) across the time course of transdifferentiation. Estimates are based on two biological replicates. Statistical comparisons were performed using a two-sided Kolmogorov–Smirnov test (not significant (n.s.), $P > 0.05$; ****$P \le 0.0001$). Black bars represent the medians, box limits are the first and third quartiles, and whiskers represent 1.5 times the interquartile range. Notches extend to 1.58 times the interquartile range divided by the square root of $n$ (~95% confidence intervals of the median values). Outliers are not shown.

We detected a decrease in CDK9 and cyclin T1 occupancies, consistent with the observed decrease in Pol II productive elongation toward 96 h (Extended Data Fig. 2c,d). Together, these results indicate that downregulation of the pre-B genes involves promoter-proximal Pol II regulation, which could be mediated by promoter-proximal pausing or termination or both.

**Two iMac gene sets differ in promoter-proximal regulation**
We next analyzed the transcription kinetics of the iMac genes ($n = 416$; Fig. 1f), which were upregulated during transdifferentiation. Metagene analysis showed an increase in TT-seq and mNET-seq signals throughout the gene from 0 to 96 h after transdifferentiation induction (Fig. 3a). Accordingly, $I$ (median $FC_{0-96} = 5.2$, $P < 2.2 \times 10^{-16}$) and promoter-proximal

occupancy of Pol II (median $FC_{0-96} = 4.5$, $P < 2.2 \times 10^{-16}$) increased significantly from 0 to 96 h (Fig. 3b, top left and middle). However, $d$ did not change significantly between the terminal time points of transdifferentiation (median $FC_{0-96} = 0.87$, $P = 0.06$) (Fig. 3b, top right). We, therefore, used $k$-means clustering ($k = 2$) with respect to $I$ and $d$ for the iMac genes and obtained two distinct gene sets with different dynamics of kinetic parameters across the transdifferentiation time course (Fig. 3b). Clustering with $k > 2$ resulted in additional gene sets with similar patterns of $I$ and $d$ changes, indicating the presence of only two major different kinetic scenarios (Extended Data Fig. 3a).

We named the obtained gene sets iMac I and iMac II, which contained 193 and 223 genes, respectively. To investigate whether the iMac gene sets differ in their biological functions, we performed GO

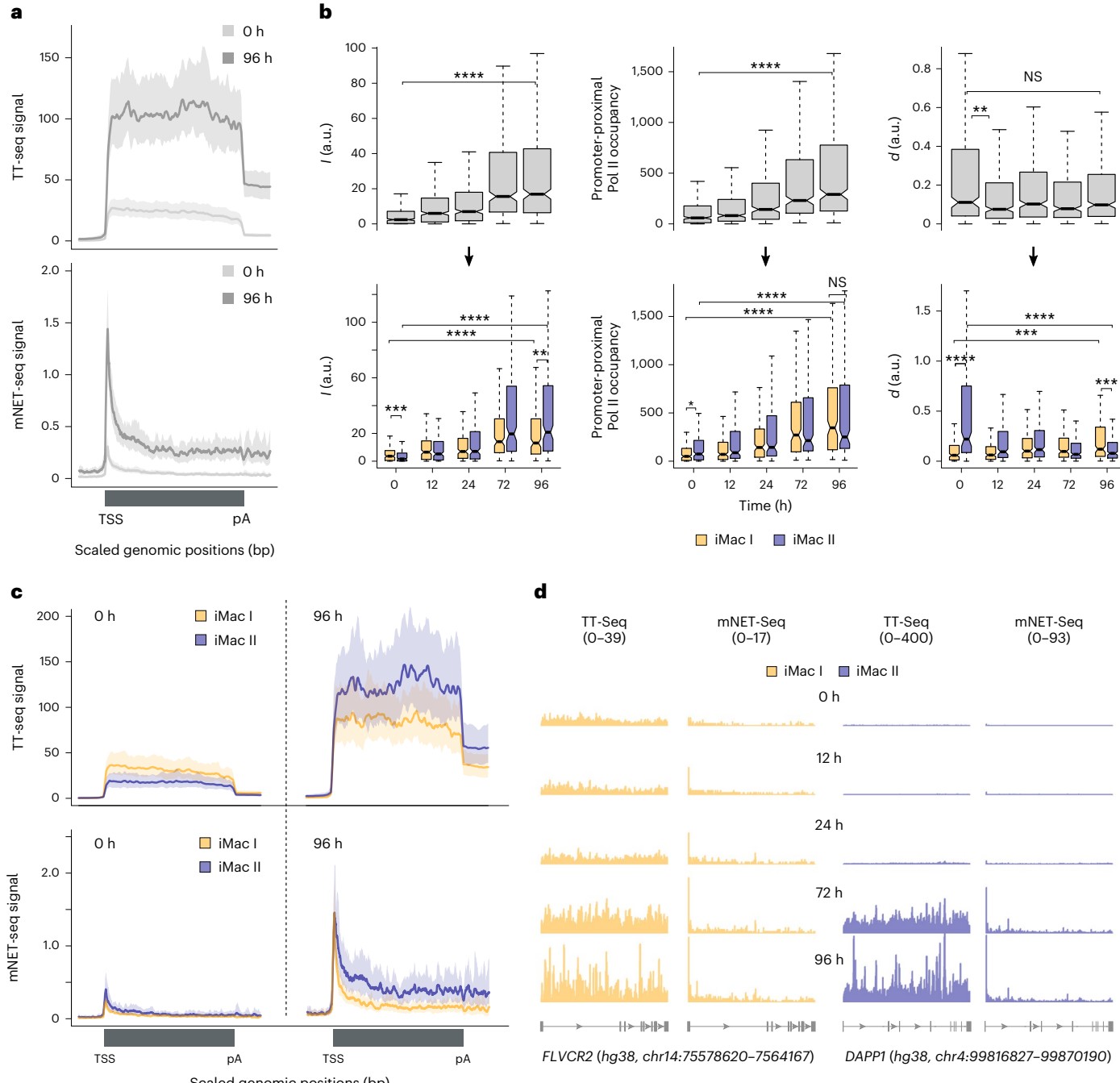

**Fig. 3 | Two iMac gene sets differ in function and promoter-proximal regulation. a**, Metagene profiles comparing TT-seq and mNET-seq signals between 0 h (light gray) and 96 h (dark gray) after transdifferentiation induction for iMac genes. Both TT-seq and mNET-seq signals were averaged for 416 iMac genes and scaled between the TSS and poly(A)-site. Data from two biological replicates are merged for illustration. Solid lines represent the averaged signal and shaded regions show the 95% confidence interval of the mean. **b**, Top: box plots showing estimates of kinetic parameters $I$ (left) and $d$ (right) and promoter-proximal Pol II occupancy (middle) for iMac genes ($n = 416$) across the time course of transdifferentiation. Bottom: box plots comparing the estimates of kinetic parameters $I$ (left) and $d$ (right) and promoter-proximal Pol II occupancy (middle) between 193 iMac I and 223 iMac II genes across the time course of transdifferentiation. Estimates are based on two biological replicates. Statistical comparisons were performed using a two-sided Kolmogorov–Smirnov test

(n.s., $P > 0.05$; *$P \leq 0.05$, **$P \leq 0.01$, ***$P \leq 0.001$ and ****$P \leq 0.0001$). Black bars represent the medians, box limits are the first and third quartiles, and whiskers represent 1.5 times the interquartile range. Notches extend to 1.58 times the interquartile range divided by the square root of $n$ (~95% confidence intervals of the median values). Outliers are not shown. **c**, Metagene profiles comparing TT-seq and mNET-seq signals for iMac I and iMac II gene sets at 0 and 96 h after transdifferentiation induction. Both TT-seq and mNET-seq signals were averaged for 193 iMac I and 223 iMac II genes and scaled between the TSS and poly(A)-site. Data from two biological replicates are merged for illustration. Solid lines represent the averaged signal and shaded regions show the 95% confidence interval of the mean. **d**, Representative examples of iMac I (*FLVCR2*) and iMac II (*DAPP1*) genes. TT-seq and mNET-seq data collected at 0, 12, 24, 72 and 96 h after transdifferentiation induction are shown. Data from two biological replicates are merged for illustration.

analysis and obtained a higher number of enriched GO terms and macrophage-specific immune functions for iMac II genes compared to iMac I (Extended Data Fig. 3c,e,f). Additionally, we performed pathway analysis using the STRING database[56] and discovered a greater number of interactions in the iMac II gene set (average node degree = 4.21) compared to iMac I (average node degree = 1.65). STRING reactome pathway analysis for iMac II revealed a significant enrichment in macrophage-related signaling cascades[57], whereas a minor enrichment in general immune response pathways was observed for iMac I (Supplementary Tables 2 and 3). This suggests that the distinct transcription kinetics of the iMac gene sets are associated with different biological functions.

iMac II genes showed a greater $I$ upregulation during transdifferentiation (median $FC_{0-96} = 11.07, P < 2.2 \times 10^{-16}$) than iMac I (median $FC_{0-96} = 2.72, P < 2.2 \times 10^{-16}$) (Fig. 3b, bottom left), which was reflected in the TT-seq profiles (Fig. 3c,d and Extended Data Fig. 3b). We observed a decrease in $d$ (median $FC_{0-96} = 0.38, P = 8.37 \times 10^{-11}$) for the iMac II genes, whereas $d$ for the iMac I genes exhibited an opposite pattern (median $FC_{0-96} = 1.94, P = 8.8 \times 10^{-5}$) (Fig. 3b, bottom right). At 0 h, $d$ was significantly higher for iMac II genes compared to iMac I genes. This can be explained by higher promoter-proximal Pol II occupancy and/or lower $I$ for iMac II genes at 0 h (Fig. 3b, bottom, and Extended Data Fig. 3b). Toward 96 h of transdifferentiation, both gene sets had similar levels of promoter-proximal Pol II occupancy (Fig. 3b, bottom middle) but the iMac I gene set showed lower mNET-seq signal in the gene body in comparison to iMac II (Fig. 3c,d). Together, this indicates a similar gain of transcriptionally engaged Pol II in the promoter-proximal region for both gene sets, yet more Pol II productive elongation for iMac II compared to iMac I toward 96 h of transdifferentiation. Notably, the occupancy of P-TEFb subunits increased for iMac genes during transdifferentiation but did not reflect the observed differences in the transcription kinetics of the iMac gene sets (Extended Data Fig. 3d). Taken together, our results indicate that upregulation of iMac genes occurs by two different mechanisms of promoter-proximal Pol II transcription regulation.

### Estimation of promoter-proximal Pol II half-life

These results raise the question whether the two different types of regulation of iMac genes in the promoter-proximal region (Fig. 3) are achieved by different mechanisms. Assuming that termination of Pol II in the promoter-proximal region is rare, the decrease in $d$ for iMac II genes (Fig. 3b, bottom right) implies a decrease in promoter-proximal pausing. Similarly, for the downregulated pre-B genes, the observed increase in $d$ and corresponding Pol II retention in the promoter-proximal region (Fig. 2) can be interpreted as an increase in promoter-proximal pausing. However, Pol II occupancy peaks in the promoter-proximal region can result from Pol II undergoing pausing or termination[38] and our mNET-seq data do not allow us to distinguish between the two. Therefore, we needed to extend our multiomics approach with a third experimental method to account for and distinguish between termination and pausing of Pol II in the promoter-proximal region. We, therefore, measured Pol II stability in the promoter-proximal region by combining inhibition of transcription initiation using triptolide with high-resolution ChIP-nexus of total Pol II. We chose ChIP-nexus because it has been previously used for promoter-proximal Pol II half-life estimation[42] and it provides information on Pol II occupancy not only downstream of the TSS but also upstream where the preinitiation complex (PIC) assembles at the promoter. Triptolide is a known inhibitor of the general transcription factor TFIIH[58]. Triptolide covalently binds the XPB subunit of TFIIH, blocking its ATP-dependent DNA translocase activity, which results in clearance of the promoter-proximal region from Pol II by either productive elongation activation or termination[33,36,42,46].

We optimized the triptolide concentration and treatment time for pre-B and macrophage-like cells. Using 5 µM triptolide for up to 30 min, we did not observe a reduction in Pol II signal because

of proteasomal degradation, as has been reported for prolonged treatment[59,60] (Extended Data Fig. 4a). We then carried out ChIP-nexus experiments, treating cells at 0 and 96 h of transdifferentiation with triptolide for 6 and 30 min for two independent biological replicates (Pearson correlation coefficient = 0.94−1) (Fig. 4a and Extended Data Fig. 4b,c). Treatment with triptolide resulted in a gradual, global loss of Pol II ChIP-nexus signal in the promoter-proximal region, which coincided with a shift of the signal upstream of the TSS to the expected site of PIC formation (Fig. 4b and Extended Data Fig. 4d). This confirmed that initiation was inhibited and Pol II was lost from the promoter-proximal region.

We further estimated the half-life of promoter-proximal Pol II by fitting the Pol II occupancy measurements from the time series of triptolide treatment to an exponential decay model (Fig. 4d and Extended Data Fig. 4e). The obtained half-lives varied mainly from 5 to 10 min at both time points of transdifferentiation (Fig. 4c), in agreement with previous studies[36,46,47]. We next ranked genes on the basis of their $I$ and $d$ estimates and divided them into four quantiles. As expected, we observed a decrease in promoter-proximal Pol II half-life with increasing $I$ (Fig. 4e and Extended Data Fig. 4f). However, an increase in $d$ generally did not correlate with an increase in promoter-proximal Pol II half-life (Fig. 4f and Extended Data Fig. 4g), which depends on both promoter-proximal pausing and termination. This indicates the importance of considering promoter-proximal Pol II stability measurements when analyzing Pol II pausing kinetics.

### Estimation of promoter-proximal Pol II termination fraction

The ChIP-nexus data after transcription inhibition allowed us to estimate the rate of Pol II occupancy loss from the promoter-proximal region caused by both productive elongation activation and termination of Pol II. We defined this rate as total Pol II turnover $r$ in the promoter-proximal region (Fig. 5a and Methods). We assumed that $r$ represents the total number of Pol II that are released from the PIC per unit time, thus providing a proxy for the transcription initiation frequency $i$ at the promoter (Fig. 5a). Because $r$ depends on both the termination of promoter-proximal Pol II and its release into productive elongation, while the productive initiation frequency $I$ provides an estimate of the latter, we could derive a relative estimate of the termination fraction of Pol II in the promoter-proximal region as $1 − I/r$ (Fig. 5a and Methods). A higher value of this relative measure indicates a higher fraction of Pol II undergoing termination and a lower fraction of Pol II entering productive elongation in the promoter-proximal region.

Our model predicts that $d$ overestimates the actual pause duration for individual polymerases for genes with high levels of promoter-proximal termination. We refer to the actual pause duration as to the time that a single Pol II spends in the promoter-proximal region before entering productive elongation. Indeed, we observed that genes with high and low estimated promoter-proximal Pol II termination fractions had correspondingly high and low $d$ values (Extended Data Fig. 4h). Moreover, for genes with low levels of promoter-proximal termination, we observed a good correlation between $d$ and the half-life of promoter-proximal Pol II, corroborating our model (Extended Data Fig. 4i). With this model for estimating the relative fraction of Pol II termination in the promoter-proximal region, we delved deeper into the transcription regulation of the previously defined genes that were upregulated and downregulated during transdifferentiation.

### Reduced Pol II termination contributes to gene upregulation

With the ChIP-nexus data at hand, we first could reinvestigate the regulatory mechanisms underlying the upregulation of the iMac gene sets (Fig. 3). Similar to all analyzed genes (Fig. 4b and Extended Data Fig. 4d), metagene analysis of Pol II ChIP-nexus data revealed a global loss of the signal from the promoter-proximal region and a further shift upstream upon triptolide treatment at 0 and 96 h of transdifferentiation (Fig. 5b). We detected a significant decrease in the half-life of promoter-proximal

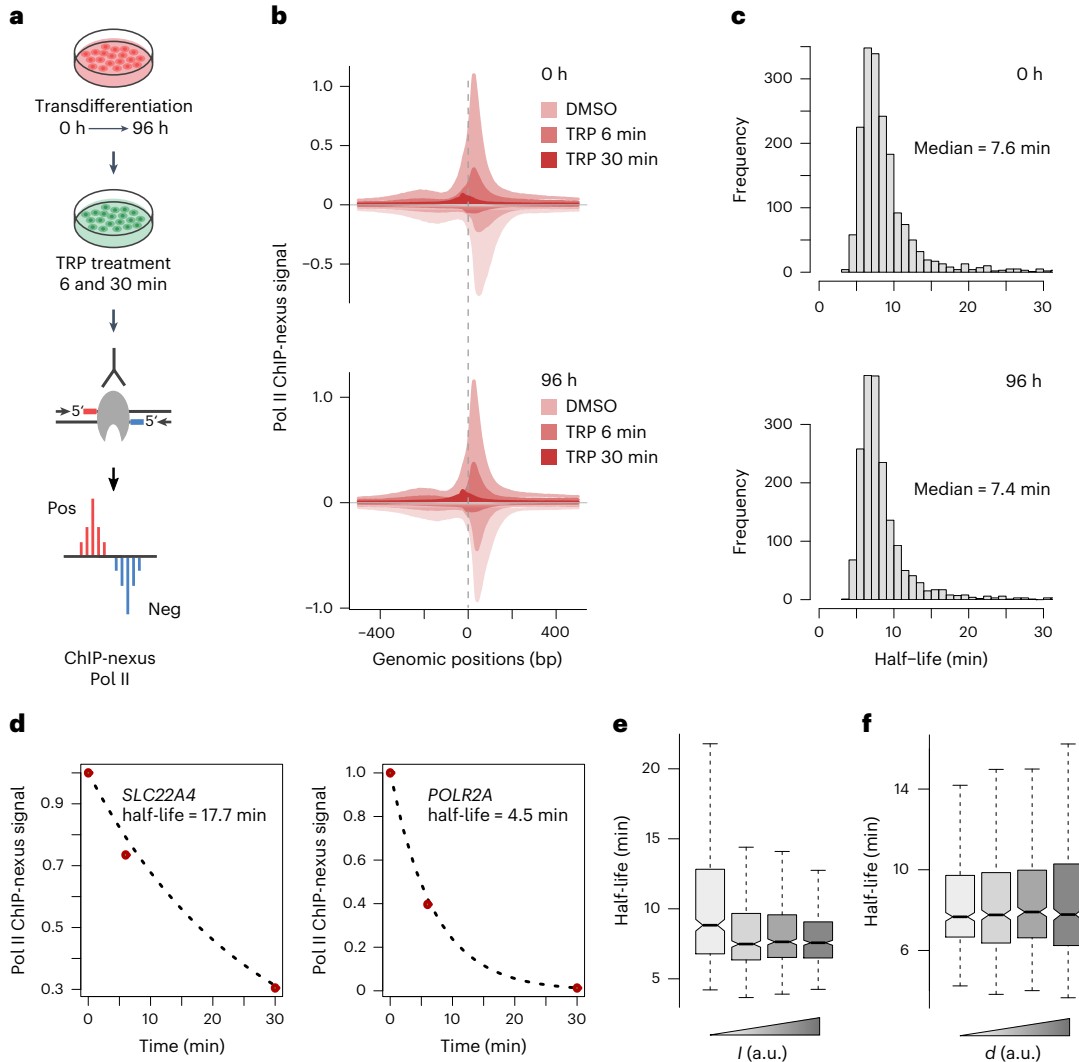

**Fig. 4 | Estimation of Pol II half-life in the promoter-proximal region.**
**a**, Schematic representation of triptolide treatment at 0 and 96 h time points of transdifferentiation followed by ChIP-nexus experiment for two independent biological replicates. TRP, triptolide. Adapted from Shao et al.[42]. **b**, Metagene profiles showing ChIP-nexus signal after 6 and 30 min of DMSO or triptolide treatment at 0 and 96 h of transdifferentiation. The positive strand is shown above the baseline (dark shade) and the negative strand is shown below the baseline (light shade). ChIP-nexus signals from two biological replicates are merged for illustration. The data are shown for 2,157 selected differentially expressed protein-coding genes (Fig. 1c). **c**, Histogram showing the estimated Pol II half-lives in the promoter-proximal region at 0 h (median = 7.6 min) and 96 h

(median = 7.4 min) of transdifferentiation ($n = 1,814$; Methods). **d**, Representative gene examples with different half-lives of promoter-proximal Pol II. The measurements of Pol II occupancy under the time course of triptolide treatment were fit to an exponential decay model (Methods). **e**,**f**, Box plots showing promoter-proximal Pol II half-lives for four quantiles of estimated $I$ (**e**) and $d$ (**f**) values ranked from lowest to highest ($n = 1,814$). Estimates are based on two biological replicates. Data are shown for 0 h after transdifferentiation induction. Black bars represent the medians, box limits are the first and third quartiles, and whiskers represent 1.5 times the interquartile range. Notches extend to 1.58 times the interquartile range divided by the square root of $n$ ($\sim$95% confidence intervals of the median values). Outliers are not shown.

Pol II for iMac II genes (median $FC_{0-96} = 0.93$, $P = 1.4 \times 10^{-3}$) but not for iMac I (median $FC_{0-96} = 0.97$) between 0 and 96 h of transdifferentiation (Fig. 5c). Consistent with the observed increase in productive initiation frequency of the iMac gene sets (Fig. 3b, left), total Pol II turnover in the promoter-proximal region increased significantly for both iMac I (median $FC_{0-96} = 2.3$, $P = 1.5 \times 10^{-10}$) and iMac II (median $FC_{0-96} = 2.92$, $P = 4.8 \times 10^{-8}$) genes, implying a significant increase in the transcription initiation frequency at the promoter (Fig. 5d). We further detected a significant decrease in the Pol II termination fraction in the promoter-proximal region for iMac II genes ($P = 1 \times 10^{-4}$) but not for iMac I ($P = 0.51$) (Fig. 5e).

Taken together, these observations suggest that generally Pol II is not stably paused in the promoter-proximal region of iMac genes at the pre-B stage. Upon transdifferentiation induction, transcriptional upregulation of iMac I genes is achieved only by increasing initiation

frequency at the promoter without significant changes in the termination fraction or half-life of Pol II in the promoter-proximal region. In contrast, iMac II genes show even greater transcriptional upregulation, driven by an increase in initiation frequency at the promoter combined with a decrease in the termination fraction of Pol II in the promoter-proximal region toward the macrophage-like stage.

### Increased Pol II termination underlies gene downregulation
Next, we investigated the regulatory mechanism underlying the downregulation of the pre-B genes (Fig. 2). We detected a similar rate of loss of pre-existing promoter-proximal Pol II under triptolide treatment at both 0 and 96 h of transdifferentiation (Figs. 4b and 5b and Extended Data Fig. 4d). Accordingly, we did not detect significant differences in promoter-proximal Pol II half-life between 0 and 96 h of transdifferentiation (median $FC_{0-96} = 0.97$) (Fig. 5c),

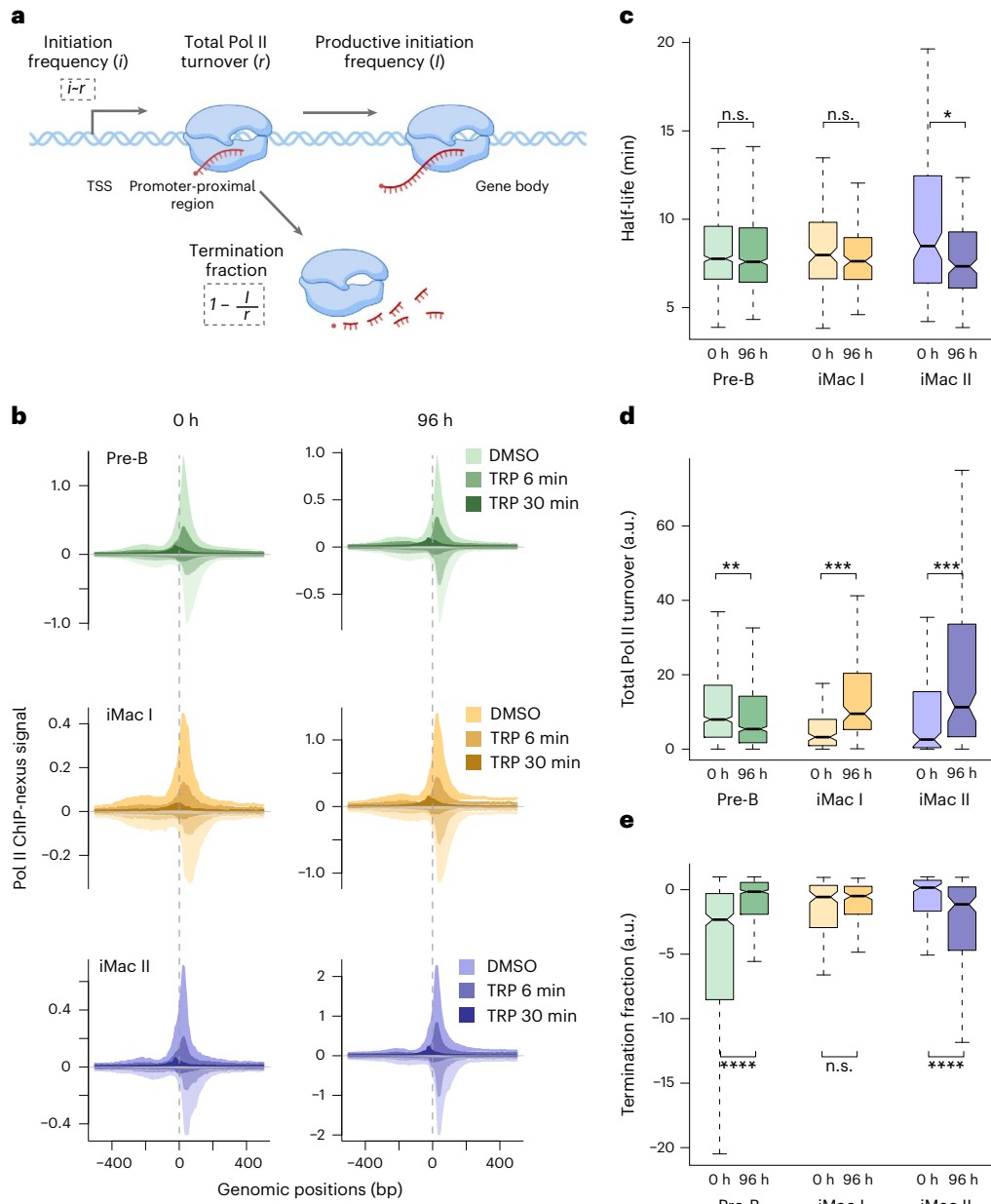

**Fig. 5 | Promoter-proximal Pol II termination contributes to both upregulation and downregulation of the genes. a**, Schematic representation of the calculation of the total Pol II turnover rate and the termination fraction in the promoter-proximal region (Methods). Panel partially created using BioRender. com. **b**, Metagene profiles showing ChIP-nexus signal after 6 and 30 min of DMSO or triptolide treatment at 0 and 96 h of transdifferentiation for 938 pre-B, 193 iMac I and 223 iMac II genes. The positive strand is shown above the baseline (dark shade) and the negative strand is shown below the baseline (light shade). ChIP-nexus data from two biological replicates are merged for illustration. **c**, Box plots comparing the promoter-proximal Pol II half-life estimates at 0 and 96 h of transdifferentiation for pre-B ($n = 833$), iMac I ($n = 146$) and iMac II ($n = 162$) genes. Estimates are based on two biological replicates. Statistical comparisons were

performed using a two-sided Kolmogorov–Smirnov test (n.s., $P > 0.05$; *$P \le 0.05$, **$P \le 0.01$, ***$P \le 0.001$ and ****$P \le 0.0001$). Black bars represent the medians, box limits are the first and third quartiles, and whiskers represent 1.5 times the interquartile range. Notches extend to 1.58 times the interquartile range divided by the square root of $n$ (~95% confidence intervals of the median values). Outliers are not shown. **d**, Box plots comparing the total Pol II turnover rate in the promoter-proximal region at 0 and 96 h of transdifferentiation for pre-B, iMac I and iMac II genes. Estimates are based on two biological replicates. Representations are as in **c**. **e**, Box plots comparing the Pol II termination fraction in the promoter-proximal region at 0 and 96 h of transdifferentiation for pre-B, iMac I and iMac II genes. Estimates are based on two biological replicates. Representations are as in **c**.

suggesting an increase in promoter-proximal Pol II termination rather than stabilization of the paused Pol II complex as a cause for the observed decrease in Pol II productive initiation frequency (Fig. 2c, left). We, thus, estimated the total Pol II turnover and the Pol II termination fraction in the promoter-proximal region (Fig. 5a and Methods). Indeed, in addition to a significant decrease in total Pol II turnover (median $FC_{0-96} = 0.76$, $P = 8.3 \times 10^{-8}$), we observed a

significant increase in the Pol II termination fraction ($P = 2.2 \times 10^{-16}$) in the promoter-proximal region for the pre-B genes from 0 to 96 h of transdifferentiation (Fig. 5d,e). Genes with a similar rate of loss of pre-existing promoter-proximal Pol II under triptolide treatment (Fig. 5b) and, thus, similar Pol II half-lives (Fig. 5c) could show differences in promoter-proximal termination (Fig. 5e), as the calculation of the Pol II termination fraction considers not only the half-life but

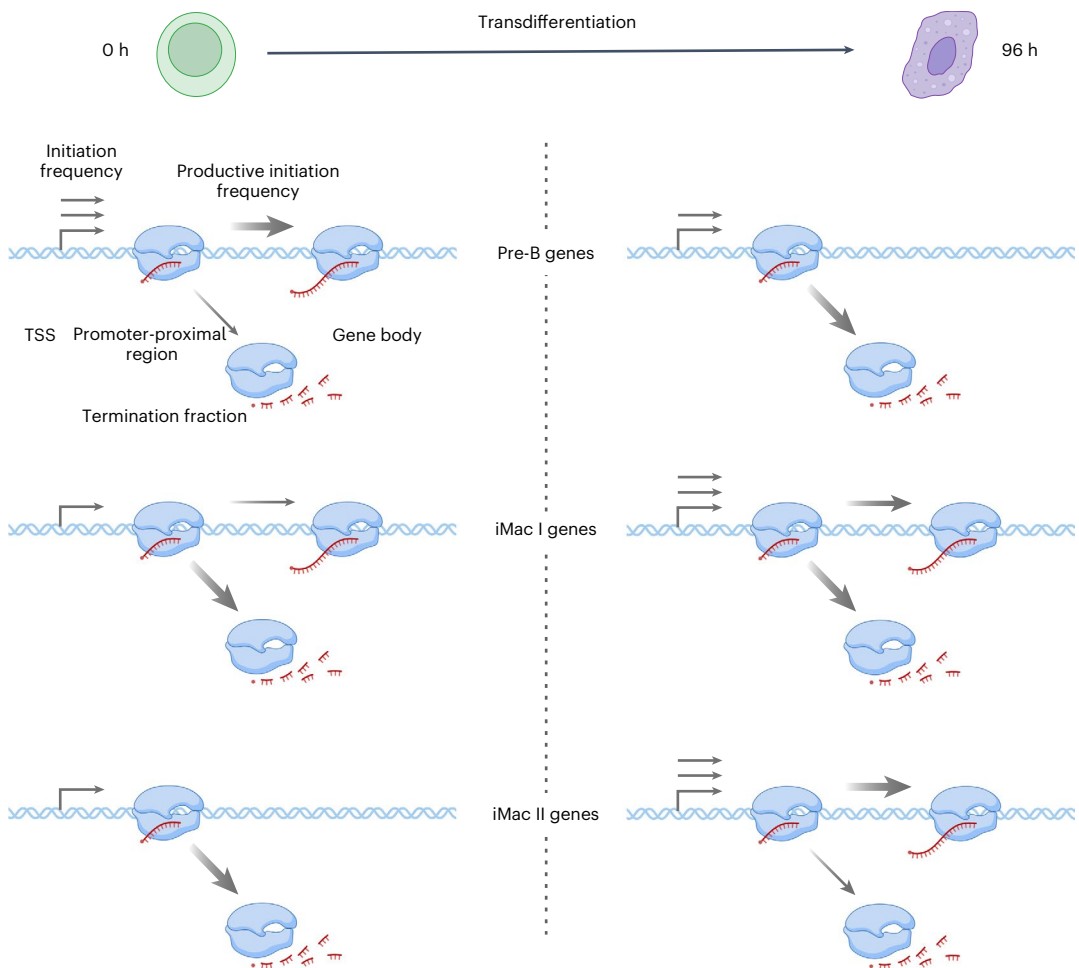

**Fig. 6 | Proposed model for promoter-proximal Pol II transcription regulation during transdifferentiation.** Distinct regulatory strategies for pre-B and iMac genes during transdifferentiation are shown. The number and thickness of the arrows represent the differences in initiation frequency and the corresponding fractions of promoter-proximal Pol II termination or productive elongation (details in text). Figure partially created using BioRender.com.

also the promoter-proximal occupancy and the productive initiation frequency of Pol II (Fig. 5a and Methods).

We, therefore, concluded that transcriptional downregulation of pre-B genes is primarily achieved by decreasing the transcription initiation frequency at the promoter together with increasing the Pol II termination fraction in the promoter-proximal region, rather than by stabilizing Pol II in the paused complex.

## Discussion

How transcription of human genes and, thus, gene activity are regulated remains a long-standing question that can only be resolved by kinetic analysis in living cells. Three processes were proposed to be targeted for the transcription regulation at the beginning of the genes: the initiation frequency, the duration of Pol II pausing and the frequency of promoter-proximal termination. We previously introduced TT-seq to monitor the productive initiation frequency, which is defined as the number of Pol II enzymes that initiate transcription, successfully pass the promoter-proximal region and enter productive elongation[39,40]. The productive initiation frequency provides the amount of full-length RNA transcripts made per time and, thus, directly measures gene activity. The productive initiation frequency can only be as high as the actual frequency of transcription initiation, which is limited by the duration of Pol II pausing in the promoter-proximal region[39,42]. Here, we extend our previous approach to the estimation of promoter-proximal termination of Pol II and show that it is also a regulated process in human cells that can contribute to both upregulation and downregulation of genes (Fig. 6).

In particular, we used a highly efficient human cell transdifferentiation model system[48–50]. In agreement with previous reports studying cell type transition processes in *Drosophila* and mammals[28,30,31,61], we detected transcriptionally engaged Pol II in the promoter-proximal region before and after the transcriptionally active state of iMac and pre-B genes, respectively (Figs. 2 and 3). The downregulation of pre-B gene transcription was mediated by both a decrease in initiation frequency and an increase in the promoter-proximal Pol II termination fraction (Figs. 5 and 6), consistent with recent findings suggesting an increase in promoter-proximal premature termination upon transcription repression stimuli in human cells[46,47]. The upregulation of iMac I genes was mediated by an increase in transcription initiation frequency alone (Figs. 5 and 6), consistent with a recent study indicating that RNA synthesis is controlled by changes in transcription initiation rather than promoter-proximal pausing during mammalian erythropoiesis[55]. Notably, we also observed a regulatory strategy for the upregulation of iMac II genes showing an increase in transcription initiation frequency with a concomitant decrease in the termination fraction and half-life of Pol II in the promoter-proximal region. This resulted in a higher transcriptional output of iMac II genes compared to iMac I genes (Figs. 5 and 6). Together with the observation of higher enrichment in macrophage-specific functions for iMac II genes (Extended Data Fig. 3c,e,f), we conclude that this promoter-proximal regulatory strategy is crucial for high expression levels of cell type determining genes upregulated during transdifferentiation. Our data befit previous studies reporting the regulatory role of promoter-proximal

Pol II pausing at target genes during cell type transition[28,30,31,61] while also allowing us to disentangle the role of promoter-proximal termination during this process.

Regulated Pol II termination in the promoter-proximal region can explain the observed discrepancies between estimates of apparent pause duration $d$ and the half-life of promoter-proximal Pol II (Fig. 4f and Extended Data Fig. 4g). Calculations of $d$ assume rare Pol II termination in the promoter-proximal region[39], resulting in $d$ values that overestimate the actual pause duration by a factor proportional to the fraction of promoter-proximal termination (Extended Data Fig. 4h,i). Thus, $d$ provides a good estimate of the actual pause duration only when promoter-proximal Pol II termination is rare. Lastly, our results show that large changes in productive transcription (Figs. 2c and 3b) are accompanied by small changes in both transcription initiation at the promoter and termination in the promoter-proximal region (Fig. 5d,e). These observations support a model in which promoter-proximal Pol II is very dynamic and a high rate of premature termination (that is, a high rate of Pol II turnover) may represent a default state[44–47].

## Online content

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

## Methods

### Cell culture

We used the previously obtained engineered human BLaER1 cell line stably expressing C/EBPα fused to the estrogen receptor hormone-binding domain and GFP[48,50]. The BLaER1 cell line is derived from precursor leukemia B cells that can be efficiently transdifferentiated into functional macrophage-like cells upon estrogen induction[48,50]. Cells were cultured in the growth medium consisting of RPMI 1640 (Thermo Fisher Scientific, 31870-074) supplemented with 10% FBS (Thermo Fisher Scientific, 10500-064), 4 mM GlutaMAX (Thermo Fisher Scientific, 35050087), 25 mM HEPES (Thermo Fisher Scientific, 15630080) and 100 U per ml penicillin–streptomycin (Thermo Fisher Scientific, 15140122) at 37 °C and 5% $CO_2$. Biological replicates were cultured independently. BLaER1 cells were regularly examined and tested negative for the *Mycoplasma* contamination using Plasmo Test *Mycoplasma* detection kit (InvivoGen, rep-pt1).

### Treatments

To induce transdifferentiation, BLaER1 cells were brought to a density of $0.4 \times 10^6$ cells per ml and mixed with 100 nM β-estradiol (Sigma-Aldrich, E2758-250MG), 10 ng $ml^{-1}$ recombinant human interleukin 3 (PeproTech, 200-03) and 10 ng $ml^{-1}$ recombinant human macrophage colony-stimulating factor (PeproTech, 300-25) in the growth medium[50]. For the 0-h control, BLaER1 cells were treated with the same concentration of solvents (ethanol and water). Cells were harvested at different time points: 0, 12, 24, 72 and 96 h after induction for RNA extraction and mNET-seq, 0, 24 and 96 h after induction for ChIP-seq and 0 and 96 h after induction for ChIP-nexus. To inhibit transcription initiation, BLaER1 cells were collected at 0 or 96 h of transdifferentiation, brought to a density of $1 \times 10^6$ cells per ml and treated with 5 μM triptolide (Sigma-Aldrich, T3652) or DMSO (Sigma-Aldrich, D2438) as a solvent control.

### Total RNA extraction and RT–qPCR

Cell pellets were resuspended in QIAzol Lysis Reagent (Qiagen, 79306) and incubated at room temperature for 5 min. Total RNA was extracted according to the manufacturer's instructions (Qiagen). To eliminate genomic DNA contamination, total RNA was treated with the TURBO DNA-free Kit (Thermo Fisher Scientific, AM1907) according to the manufacturer's instructions. Complementary DNA synthesis was performed using Maxima H Minus reverse transcriptase (Thermo Fisher Scientific, EP0753) according to the manufacturer's instructions. qPCR was conducted with SYBR Select Master Mix (Thermo Fisher Scientific, 4472919) according to the manufacturer's instructions. Primer sequences used for qPCR are listed in Supplementary Table 1.

### mNET-seq

mNET-seq was performed as described previously[52,53,62] with minor modifications. Briefly, two independent biological replicates of BLaER1 cells were subjected to transdifferentiation and collected at 0, 12, 24, 72 and 96 h after induction. Cells were further used in the amount of $2 \times 10^8$ per replicate per time point. All buffers were supplemented with protease inhibitor cocktail (Sigma-Aldrich, P8340) and phosphatase inhibitors (Millipore Sigma, 4906837001). After washing with DPBS (Thermo Fisher Scientific, 14190169), cellular fractionation of $2 \times 10^7$ cells per reaction was performed according to the previously published protocol[63]. Isolated chromatin was subjected to micrococcal nuclease (New England Biolabs (NEB), M0247S) digestion at 37 °C and 1,400 rpm for 90 s followed by stopping the reaction with 25 mM EGTA (Bioworld, 40520008-1). The solution was clarified by centrifugation at 4 °C and 13,000$g$ for 5 min and the supernatants corresponding to the same sample were pooled and used for IP. Supernatant was diluted eight-fold with IP buffer (50 mM Tris-HCl pH 7.4, 150 mM NaCl, 0.05% NP-40 and 0.3% empigen BB (Sigma-Aldrich, 30326)). RNA Pol II antibody (MBL Life science, MABI0601) was coupled to Dynabeads M-280 sheep

anti-mouse IgG (Thermo Fisher Scientific, 11201D) according to the manufacturer's instructions and added to the digested chromatin at 30 μg per $2 \times 10^8$ cells. IP was performed at 4 °C and 8 rpm on a rotating wheel for 1 h. Afterward, beads were washed seven times with IP buffer and one time with PNKT buffer (1× T4 PNK buffer (NEB, M0236L) and 0.1% Tween-20). For 5′ RNA phosphorylation, beads were resuspended in PNK reaction mix (1× T4 PNK buffer (NEB, M0236L), 0.1% Tween-20, 1 mM ATP (Cell Signaling Technology, 9804S) and T4 polynucleotide kinase (phosphatase minus) (NEB, M0236L) and incubated at 37 °C and 800 rpm for 10 min. After the reaction, beads were washed one time with IP buffer and mixed with QIAzol lysis reagent (Qiagen, 79306) by vortexing for 1 min. RNA was extracted according to the manufacturer's instructions (Qiagen). Precipitation of RNA was performed with GlycoBlue coprecipitant (Thermo Fisher Scientific, AM9515) in 100% ethanol overnight. RNA was size-selected in a range of 25–110 nt using a denaturing 6% polyacrylamide gel with 7 M urea. Then, RNA was extracted from the gel in elution buffer (1 M sodium acetate pH 5.5 and 1 mM of EDTA pH 8.0) on a rotating wheel and precipitated with GlycoBlue coprecipitant (Thermo Fisher Scientific, AM9515) in 100% ethanol overnight. RNA libraries were prepared with NEBNext Multiplex small RNA library prep set for Illumina (NEB, E7300S) according to the manufacturer's instructions. Libraries were size-selected with 4% E-Gel high-resolution agarose gels (Thermo Fisher Scientific, G501804) and purified using QIAquick gel extraction kit (Qiagen, 28706X4) according to the manufacturer's instructions. Concentration and fragment size distribution of the libraries were estimated using Fragment Analyzer (Agilent). Libraries were sequenced on Illumina NEXTseq 550 using 75 cycles paired-end mode.

### ChIP-seq

ChIP-seq protocol was performed as described[64] with minor modifications. Briefly, two biological replicates of $3 \times 10^7$ BLaER1 cells were collected at 0, 24 and 96 h after transdifferentiation induction. For double crosslinking, cells were washed once with DPBS (Thermo Fisher Scientific, 14190169) and fixed in DPBS first with 2 mM DSG (Thermo Fisher Scientific, 20593) at room temperature for 20 min and then with 1% methanol-free formaldehyde (Thermo Fisher Scientific, 28908) at room temperature for 10 min. For quenching, 125 mM glycine (Sigma-Aldrich, 50046) was added to the cells and incubated at room temperature for 5 min. Fixed cells were spun down and washed twice with ice-cold DPBS (Thermo Fisher Scientific, 14190169). All the buffers were supplemented with protease inhibitor cocktail (Sigma-Aldrich, P8340) and phosphatase inhibitors (Millipore Sigma, 4906837001). The cell pellet was resuspended in Farnham lysis buffer (5 mM PIPES pH 8.0, 85 mM KCl and 0.5% NP-40) and incubated on ice for 10 min. Isolated nuclei were washed once with ice-cold DPBS (Thermo Fisher Scientific, 14190169) and resuspended in sonication buffer (10 mM Tris-HCl pH 7.5, 1 mM EDTA and 0.4% SDS) followed by incubation on ice for 10 min. Suspension was transferred into a 1-ml AFA milliTUBE (Covaris, 520130) and subjected to sonication using S220 focused ultrasonicator (Covaris) with the following parameters: duty cycle, 5%; peak incident power, 140 W; 200 cycles per burst; processing time, 960 s; bath temperature, 4–7 °C; continuous degassing mode; water level, 8. Sonicated chromatin was centrifuged at 4 °C and 10,000$g$ for 15 min and supernatant was transferred to a new tube. DNA was quantified and analyzed on a 1% agarose gel to confirm a fragment size distribution of 200–500 bp. Antibodies were coupled to Dynabeads Protein G (Thermo Fisher Scientific, 10004D) according to the manufacturer's instructions. The following antibodies were used: anti-cyclin T1 antibody (Cell Signaling, 81464) in the amount of 12.5 μl per sample, anti-CDK9 antibody (Abcam, ab239364) in the amount of 7.9 μg per sample and *Drosophila* anti-H2Av antibody (Active Motif, 61686) in the amount of 0.5 μg per sample. IP was performed with 50 μg chromatin per sample. *Drosophila* S2 spike-in chromatin was produced as previously described[64] and added in the amount of 122 ng

per sample. A total of 1% of each sample was kept as input and stored at 4 °C. Chromatin was diluted with IP buffer (56.25 mM Tris-HCl pH 7.5, 157.5 mM NaCl, 1 mM EDTA, 1.125% Triton X-100 and 0.1125% sodium deoxycholate) to obtain 0.1–0.05% SDS concentration. Chromatin was mixed with antibody–bead complexes and incubated on a rotating wheel at 4 °C overnight. On the next day, beads were washed five times with LiCl wash buffer (100 mM Tris-HCl pH 7.5, 500 mM LiCl, 1% NP-40 and 1% sodium deoxycholate) and one time with TE buffer (10 mM Tris-HCl pH 8.0 and 1 mM EDTA). DNA was eluted from the beads at 70 °C for 10 min and decrosslinked at 65 °C overnight along with input samples. DNA was subsequently treated with RNase A (Thermo Fisher Scientific, EN0531) and proteinase K (Thermo Fisher Scientific, AM2546) followed by precipitation in 100% ethanol. DNA concentration was measured with a Qubit 2.0 fluorometer (Thermo Fisher Scientific) and the IP enrichment over input was analyzed with RT–qPCR. Equal amounts of DNA were used for the library preparation with NEBNext Ultra II DNA library prep kit for Illumina (NEB, E7645S) according to the manufacturer's instructions. Concentration and fragment size distribution of the libraries were estimated using Fragment Analyzer (Agilent). Libraries were sequenced on Illumina NEXTseq 550 using 75 cycles in paired-end mode.

### ChIP-nexus

Two biological replicates of BLaER1 cells were collected at 0 and 96 h after transdifferentiation induction and treated with 5 µM triptolide (Sigma-Aldrich, T3652) or DMSO (Sigma-Aldrich, D2438) as described in Treatments section. For single crosslinking, 3e7 cells per condition were fixed in the growth media with 1% methanol-free formaldehyde (Thermo Fisher Scientific, 28908) at room temperature for 10 min. Cell lysis, nuclei isolation, chromatin shearing and quantification, IP were performed as described in ChIP-seq section. IP was performed with 60 µg chromatin per sample. *Drosophila* S2 spike-in chromatin was added in the amount of 244 ng per sample. For IP, the following antibodies were used: RNA Pol II NTD antibody (Cell Signaling, 14958) in the amount of 12 µl per sample, *Drosophila* H2Av antibody (Active Motif, 61686) in the amount of 1 µg per sample. After IP, all downstream steps were performed as described[65] with minor modifications listed below. Amplified DNA libraries were size-selected using 4% E-Gel High-ReSolution Agarose Gels (Thermo Fisher Scientific, G501804) and purified using QIAquick Gel Extraction Kit (Qiagen, 28706×4) according to the manufacturer's instructions. Concentration and fragment size distribution of the libraries were estimated using Fragment Analyzer (Agilent). Libraries were sequenced on Illumina NEXTseq 550 using 75 cycles paired-end mode.

### Western blotting of the whole cell lysate

Two biological replicates of BLaER1 cells were collected at 0 and 96 h of transdifferentiation and treated with triptolide (Sigma-Aldrich, T3652) or DMSO (Sigma-Aldrich, D2438) as described above. A total of $3 \times 10^6$ cells per each condition were collected by centrifugation at room temperature and 300$g$ for 5 min. The cell pellet was resuspended in radioimmunoprecipitation assay buffer (50 mM Tris-HCl pH 8.0, 150 mM NaCl, 1% NP-40, 0.5% sodium deoxycholate and 0.1% SDS) supplemented with 500 U per ml benzonase (Sigma-Aldrich, E1014), 2 mM MgCl$_2$, protease inhibitor cocktail (Sigma-Aldrich, P8340) and phosphatase inhibitors (Millipore Sigma, 4906837001). The lysate was incubated on ice for 20 min with occasional mixing and centrifuged at 4 °C and 21,123$g$ for 15 min. The supernatant was transferred to a new tube and protein concentration was measured using Bradford assay (Bio-Rad, 5000006) according to the manufacturer's instructions. A total of 7–10 µg of the protein was loaded in NuPAGE LDS sample buffer (Thermo Fisher Scientific, NP0007) supplemented with 400 mM DTT and subjected to SDS–PAGE (Bio-Rad) followed by the transfer to the PVDF membrane (Bio-Rad, 1704156). The membrane was blocked using 5% milk in PBS containing 0.05% Tween-20

(Sigma-Aldrich, P1379) and incubated using 2% milk in PBS containing 0.05% Tween-20 (Sigma-Aldrich, P1379) with the following primary antibodies: anti-RNA Pol II NTD antibody (Santa-cruz, sc-55492; dilution 1:200) and anti-GAPDH antibody (Sigma-Aldrich, G8795; dilution 1:20,000). Next, the membranes were washed in PBS containing 0.05% Tween-20 (Sigma-Aldrich, P1379) and incubated with horseradish peroxidase-coupled secondary anti-mouse antibody (Abcam, ab5870; dilution 1:3,000). After washing in PBS containing 0.05% Tween-20 (Sigma-Aldrich, P1379), the membranes were developed with Pierce ECL Plus western blotting substrate (Thermo Scientific, 32109) on an INTAS imager according to the manufacturer's instructions.

### Major isoform annotation

Salmon version 1.3.0 (ref. [66]) was used to quantify the counts for each isoform of a gene and to select the major isoforms from our RNA-seq dataset. An isoform of a gene from the GENCODE v24 GRCh38.p5 annotation was defined as major if it was present in an amount greater than 70% of the total mean transcripts per million for at least one of the time points in our analysis (0, 12, 24, 72 and 96 h after transdifferentiation induction) and if no other isoform of the same gene had this property at any other time point. Major isoforms for genes on chromosome M were discarded from further analysis. The final annotation contained 8,765 protein-coding genes with major isoforms called.

### TT-seq data processing and normalization

TT-seq BAM files[50] were processed in the R/Bioconductor environment. Read pairs were discarded from further analysis if they spanned a region other than the major isoforms plus 500 bases upstream and downstream. Expressed genes were defined as those having ten reads per kilobase mapped to them at at least one of the time points of data collection (0, 12, 24, 72 and 96 h after transdifferentiation induction). Read counts were generated using custom R scripts and corrected for antisense bias (ratio of spurious reads originating from the opposite strand introduced by the RT reactions) using antisense bias ratios obtained from positions in regions without antisense annotation with a coverage of at least 100 according to the defined major isoforms. The DESeq2 algorithm[67] was used to calculate size factors to normalize the data used for all further analysis and to perform differential expression analysis.

### Estimation of productive initiation frequency

Productive initiation frequency $I$ was estimated in a similar way as described previously[39,40] with minor modifications. To avoid bias from using different number of cells for different time points, we estimated $I$ independent of the number of cells for our dataset. For each gene $g$, the productive initiation frequency $I_g$ (estimated in arbitrary units (a.u.)) was calculated as

$$I_g = \frac{\text{cov}_g}{L_g}$$

with TT-seq coverage $\text{cov}_g$ and length $L_g$. Note that $\text{cov}_g$ and $L_g$ were restricted to nonfirst exons for multiexon genes and to 300 bp downstream of the TSS to pA for single-exon genes.

### mNET-seq data processing and normalization

Paired-end reads of 75-bp length were collected for the samples. Quality check was performed using FastQC[68]. Reads were mapped to the human genome (GRCh38) using STAR aligner[69]. Further data processing was performed in the R/Bioconductor environment using custom scripts. To identify transcriptionally engaged Pol II positions, we took the last incorporated base (3′ end of the RNA), which is the first mapped base in read 2, and used only this position in downstream analyses. Counts were calculated for the annotated genes using a custom R script for regions of interest depending on the analysis. From the mNET-seq data,

we calculated and corrected for antisense bias as described previously[51,64]. DESeq2 (ref. [67]) size factors were calculated from the gene counts and used for normalization.

### Detection of promoter-proximal pause sites

mNET-seq data were used to determine promoter-proximal Pol II occupancy peaks for the annotated genes. A gene was selected for downstream analysis if there was a clear maximum in the mNET-seq signal profile within the first 250 bp downstream of the TSS (that is, the maximum value had to be at least five times greater than the median of the nonzero values in this window). Such an mNET-seq peak was identified for 4,560 genes at at least one time point during transdifferentiation. Bias because of variations in the promoter-proximal peak position was reduced by further limiting the analysis to only those genes for which the peak position did not change substantially during transdifferentiation (s.d. between time points ≤ 75 bp). This resulted in a total of 4,309 genes with well-defined mNET-seq peaks in the promoter-proximal region, which were further used for the calculation of kinetic parameters.

### Estimation of apparent pause duration

Apparent pause duration $d$ of a gene $g$ was defined as the ratio of mNET-seq signal within a window of 200 bp around the promoter-proximal mNET-seq peak positions (described above; pause window coverage, PWcov) to the respective $I$ as described previously[39,40]. Contrary to the previously described approach, we did not use an additional normalization factor and estimations are, hence, in a.u. After removing the genes that had no TT-seq signal at one or more time points from the data, $d$ was calculated for 2,157 DE protein-coding genes at all time points and this subset was used for further analysis related to promoter-proximal transcription regulation.

$$d_g = \frac{\text{PWcov}_g}{I_g}$$

### Classification and clustering of genes

Genes were classified into four groups on the basis of their RNA synthesis changes. The upregulated and downregulated genes were identified by checking whether the TT-seq coverage was maximum or minimum at 0 h and 72 or 96 h, respectively. To identify differences in pause regulation of the upregulated iMac genes, we further clustered them on the basis of $I$ and $d$ together ($k = 2$) using a bootstrapped $k$-means clustering algorithm. To minimize bias, all clusterings were performed with multiple values of $k$ before settling on those described in the results.

### GO and protein–protein interaction analysis

GO analysis for groups of genes was performed using DAVID[70]. GOTERM_BP_1 data from DAVID was used for the analysis, and the plots were generated using a custom R script. To investigate the interactions between proteins encoded by different sets of genes, we used the multiple protein functionality of the STRING database[56]. It was also used to generate the interaction networks, subcellular localization and reactome pathways.

### ChIP-seq data processing and normalization

Paired-end reads of 75-bp length were collected for the samples. Reads were quality-checked using FastQC[68]. Reads were mapped to the human genome (GRCh38) using the Bowtie2 aligner[71] with default parameters. Further data processing was performed in the R/Bioconductor environment using custom scripts. Duplicate reads were defined as those with the same start and end of mapped fragments and were discarded from the data. Reads with insert sizes greater than 500 bp were also discarded and the remaining reads were converted to run-length encoding (RLE) lists. Counts were calculated for the annotated genes using a custom R script for regions of interest. DESeq2 (ref. [67]) size factors were calculated from the counts at the regions of interest and used for normalization.

### ChIP-nexus data processing and normalization

ChIP-nexus data were processed as described previously[72] with minor modifications. Reads were checked for adaptor content using CutAdapt 2.3 (ref. [73]) and any regions with adaptors were removed. Furthermore, to remove the region with barcodes, 9 bp from the 5′ ends of the reads were trimmed. Reads were then mapped using Bowtie2 (ref. [71]) with default parameters to a combined human and *Drosophila* genome (dm6, GCF_000001215.4) and the samples showed an average of 74% mapping efficiency, consistent with data from the original ChIP-nexus publications[42,72]. Further data processing was performed as described previously[72]. Duplicates were removed on the basis of mapping locations and mapped reads were further trimmed to their 3′ end to extract the position of Pol II, before saving as RLE lists for further analysis. *Drosophila* spike-ins were used for normalization as described.

### Estimation of promoter-proximal Pol II half-life

Pol II ChIP-nexus coverages at identified peaks ±20 bp were used to estimate the half-life of Pol II in the promoter-proximal region. Genes with zero ChIP-nexus coverage in DMSO control or triptolide-treated samples were excluded from the half-life estimation, reducing the number of DE protein-coding genes from 2,157 to 1,877. In addition, the few genes with an observed increase in ChIP-nexus coverage in triptolide-treated samples were also excluded, leaving a final set of 1,814 genes for downstream analysis (pre-B, $n = 833$; iMac I, $n = 146$; iMac II, $n = 162$). For these genes, the normalized coverages for DMSO control and triptolide-treated samples were fit to an exponential decay model to estimate the decay constant ($k$) as described previously[42]. Half-lives were then calculated from the decay constant as $\ln 2/k$.

### Estimation of total turnover rate and termination fraction

Exponential fitting of the ChIP-nexus data can be used to estimate the rates of eviction of the polymerase from the promoter-proximal region. The half-life ($t_{1/2}$) and the Pol II signal in the promoter-proximal region ($p_0$) at a steady state provide the total Pol II turnover rate $r$:

$$r = \frac{dp}{dt} = -\ln 2 \times \frac{p_0}{t_{1/2}}$$

TT-seq data provide an estimate of the rate of promoter-proximal Pol II release into productive elongation in the form of productive initiation frequency $I$. If all the terms are exact and the calculations of $r$ and $I$ are free of bias, the promoter-proximal termination rate would be $|r| − I$. However, because this cannot be assumed, a useful quantity to measure the relative differences in the productive elongation fraction (proportion of initiated Pol II released into elongation) is $I/|r|$. A greater $I/|r|$ for a gene indicates that more of its promoter-proximal Pol II is released into productive elongation and vice versa. An estimate of the termination fraction in the promoter-proximal region can then be derived as $1 − I/|r|$.

$$\text{Productive elongation fraction} = \frac{I}{|r|}$$

$$\text{Termination fraction} = 1 - \frac{I}{|r|}$$

This quantity gives a relative estimate of the promoter-proximal termination fraction up to a proportionality constant.

### Statistics and reproducibility

All comparisons were performed using the Kolmogorov–Smirnov test in R. No statistical method was used to predetermine sample size. No data were excluded from the analyses. The experiments were not randomized. The investigators were not blinded to allocation during experiments and outcome assessment.

# Article

**Reporting summary**

Further information on research design is available in the Nature Portfolio Reporting Summary linked to this article.

## Data availability

Next-generation sequencing datasets generated in this study are available for download from the Gene Expression Omnibus (GEO) under accession code GSE235181. Published TT-seq data[50] used in this study are available for download from the GEO under accession code GSE131620. Supplementary Data 1 contains transcription kinetics data for protein-coding major isoforms. Source data are provided with this paper.

## Code availability

Data analysis scripts generated in this study are available from Zenodo (https://doi.org/10.5281/zenodo.14361018)[74].

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

## Acknowledgements

We thank J. Choi for the help with the initial mNET-seq data analysis and providing TT-seq data before publication. We thank A. Fischer for the help with ChIP-seq experiments. We thank K. Maier and P. Rus for the help with fragment analyzer and sequencing. We thank L. Caizzi and T. Velychko for providing spike-ins for ChIP-seq and ChIP-nexus experiments. We thank all past and current members of the P.C. laboratory for the scientific discussions, especially, K. Žumer, L. Caizzi, A. Sawicka and T. Velychko. We thank A. Tresch for the scientific discussions. K.L. was supported by the International Max Planck Research School for Molecular Biology and A.D. was supported by the International Max Planck Research School for Genome Science; both are part of the Göttingen Graduate School for Neurosciences, Biophysics and Molecular Biosciences. P.C. was supported by the Max Planck Society.

## Author contributions

K.L., A.D., M.L. and P.C. conceptualized and planned the project. K.L. designed and performed the experiments. A.D. designed and conducted the bioinformatic analysis. K.L., A.D. and M.L. interpreted the data. B.S. and M.L. advised on the bioinformatic analysis. K.L. and A.D. wrote the original draft of the paper with input from all authors. P.C. and M.L. reviewed and edited the paper. K.L. and A.D. visualized the data. P.C. and M.L. supervised the project. P.C. acquired funding.

## Funding

## Competing interests

The authors declare no competing interests.

## Additional information

**Extended data** is available for this paper at https://doi.org/10.1038/s41594-025-01486-9.

**Correspondence and requests for materials** should be addressed to Michael Lidschreiber or Patrick Cramer.

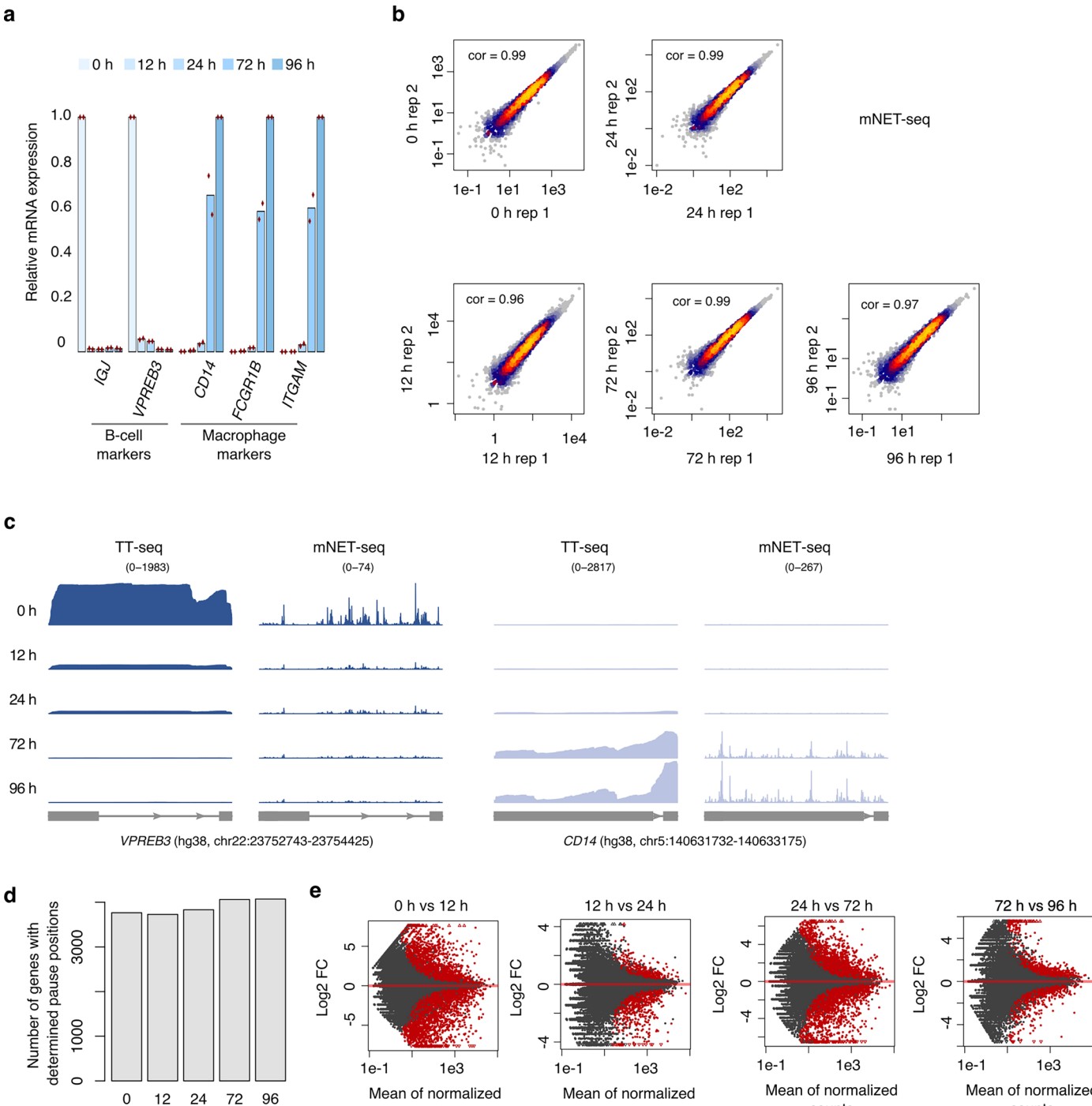

**Extended Data Fig. 1 | Multi-omics analysis of human cell type transition.**
**a**) RT-qPCR results showing the relative mRNA expression of B-cell and macrophage markers during the transdifferentiation time course (0, 12, 24, 72 and 96 h after induction). The signal was normalized to GAPDH expression. The data points correspond to two independent biological replicates for each sample. **b**) Pearson correlation between two independent biological replicates of mNET-seq data collected at 0, 12, 24, 72 and 96 h after transdifferentiation induction. A heat color gradient indicates the density of overlapping points based on a Kernel Density Estimation done by the 'heatscatter' function

from the 'LSD' R package. Areas with the highest density are shown in yellow. **c**) Representative examples of the B-cell (*VPREB3*) and macrophage (*CD14*) markers TT-seq and mNET-seq data collected at 0, 12, 24, 72 and 96 h after transdifferentiation induction. TT-seq and mNET-seq data from two biological replicates were merged for illustration. **d**) Number of annotated protein-coding genes with called paused Pol II peak during the transdifferentiation time course (0, 12, 24, 72 and 96 h after induction). **e**) MA plots showing differentially expressed genes ($n = 31{,}709$) based on TT-seq during transdifferentiation. Significantly up- or downregulated genes are colored in red (padj < 0.05).

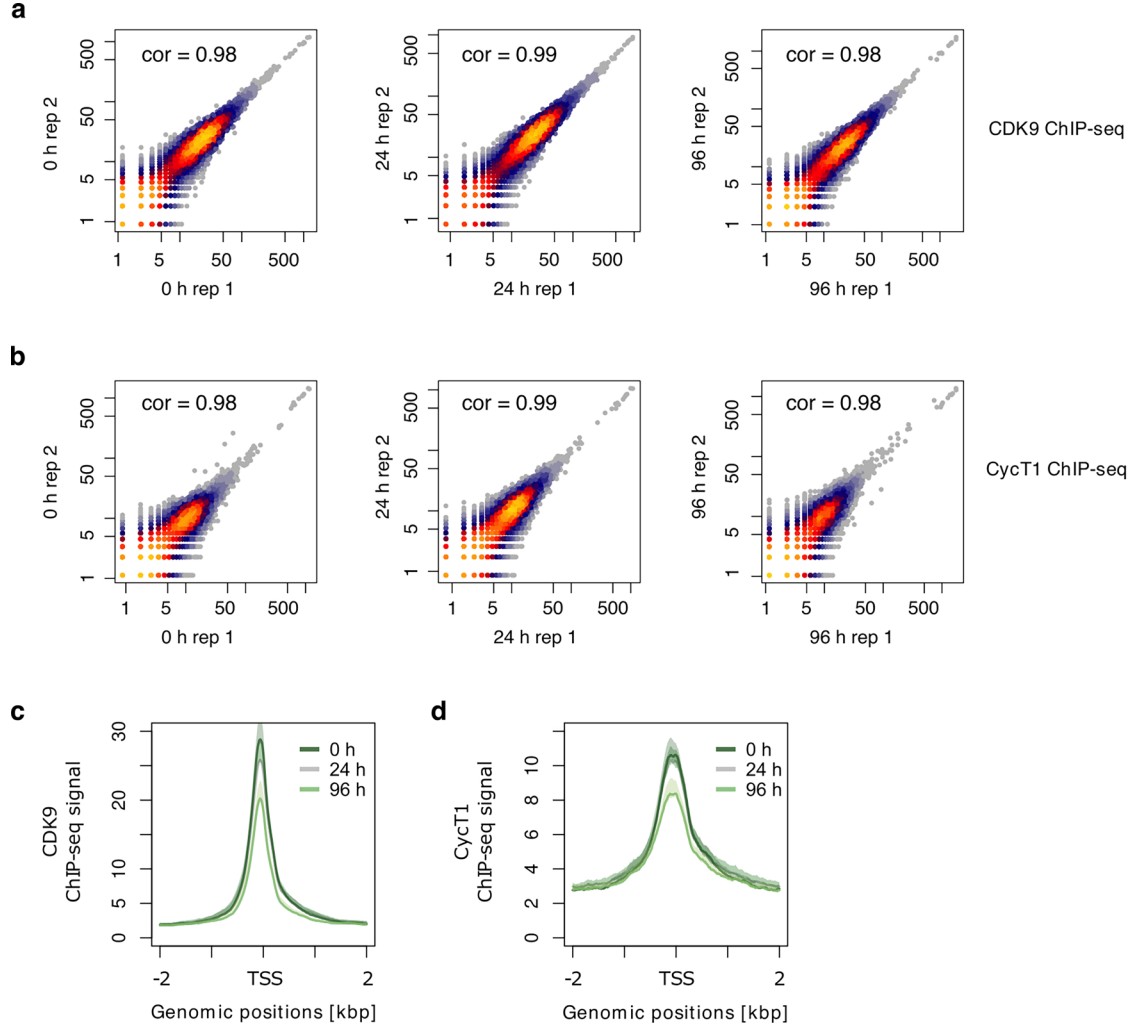

**Extended Data Fig. 2 | Occupancy of P-TEFb subunits decreases during transdifferentiation for pre-B genes. a**) and **b**) Pearson correlation between two independent biological replicates of the CDK9 (**a**) or cyclin T1 (**b**) ChIP-seq data collected at 0, 24 and 96 h after transdifferentiation induction. Heat color gradient as in Extended Data Fig. 1b. **c**) and **d**) Metagene profiles showing CDK9

(**c**) or cyclin T1 (**d**) ChIP-seq signal for 938 pre-B genes. Data is shown for 0, 24 and 96 h of transdifferentiation. ChIP-seq data from two biological replicates were merged for illustration. Solid lines represent the averaged signal, shaded regions show the 95% confidence interval of the mean.

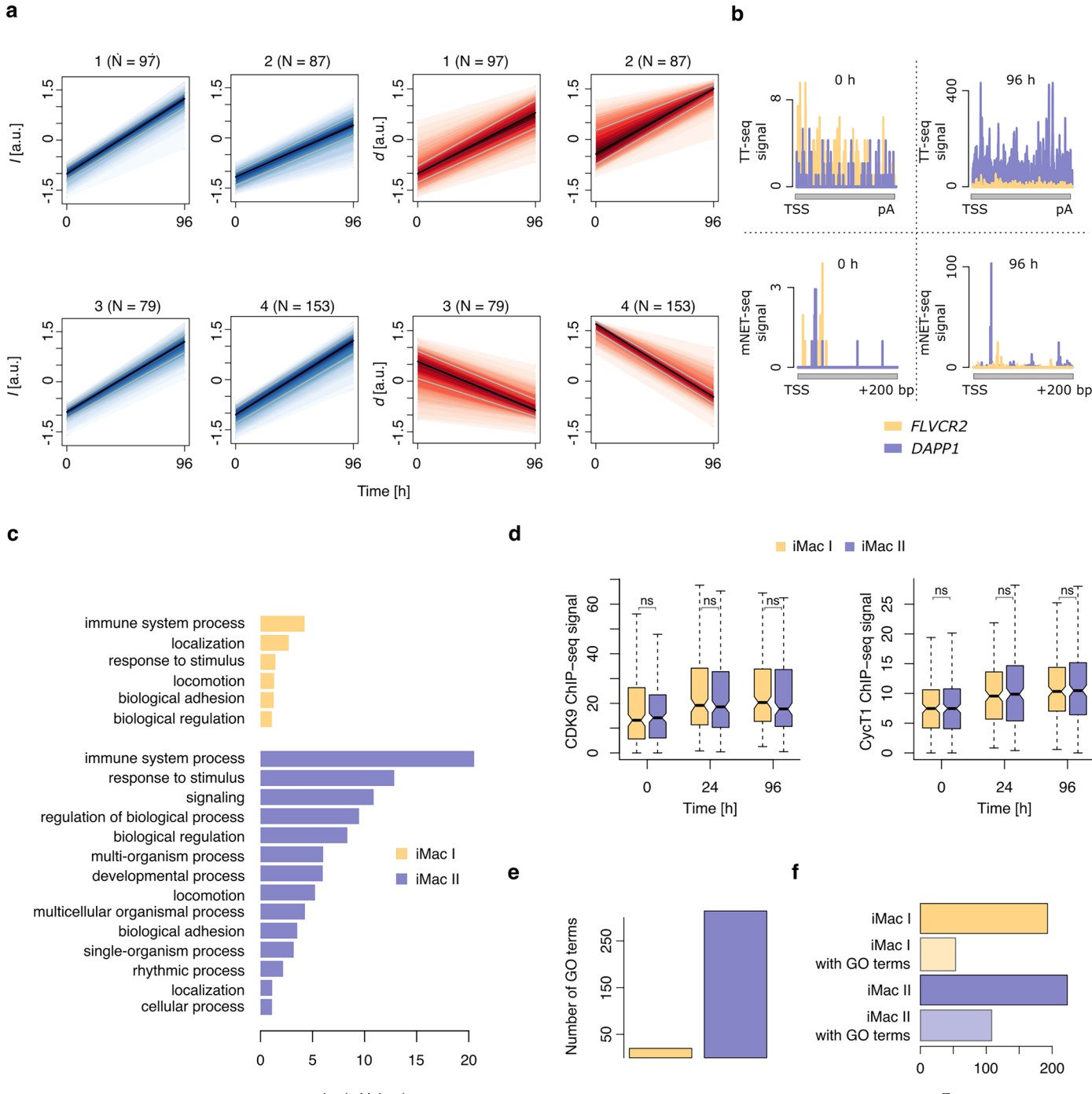

**Extended Data Fig. 3 | Two iMac gene sets differ in function and promoter-proximal regulation. a**) Representation of the k-means ($n = 4$) clustering of the kinetic parameters $I$ and $d$ for 416 iMac genes. **b**) Representative examples of iMac I (*FLVCR2*) and iMac II (*DAPP1*) genes. TT-seq (top) and mNET-seq (bottom) data collected at 0 (left) and 96 (right) h after transdifferentiation induction are shown. Data from two biological replicates were merged for illustration. mNET-seq data is shown only in the promoter-proximal region from TSS to TSS + 200 bp. $d$ values for *FLVCR2* and *DAPP1* are 0.33 and 0.84 at 0 h and 1.17 and 0.11 at 96 h, respectively. **c**) Gene ontology analysis results for iMac I and iMac II gene sets. **d**) Box plots comparing CDK9 and cyclin T1 ChIP-seq signal

between 193 iMac I and 223 iMac II genes during transdifferentiation. Data from two biological replicates were merged. Statistical comparisons were performed using the two-sided Kolmogorov-Smirnov test (ns, $p > 0.05$; *$p \leq 0.05$; **$p \leq 0.01$; ***$p \leq 0.001$; ****$p \leq 0.0001$). Black bars represent medians, box limits are the first and third quartiles, and whiskers represent 1.5 times the interquartile range. Notches extend to 1.58 times the interquartile range divided by the square root of $n$ (~95% confidence intervals of the median values). Outliers not shown. **e**) Comparison of gene ontology term numbers enriched in the iMac I ($n = 193$) and iMac II ($n = 223$) gene sets. **f**) Representation of the iMac I and iMac II gene fractions enriched in gene ontology terms.

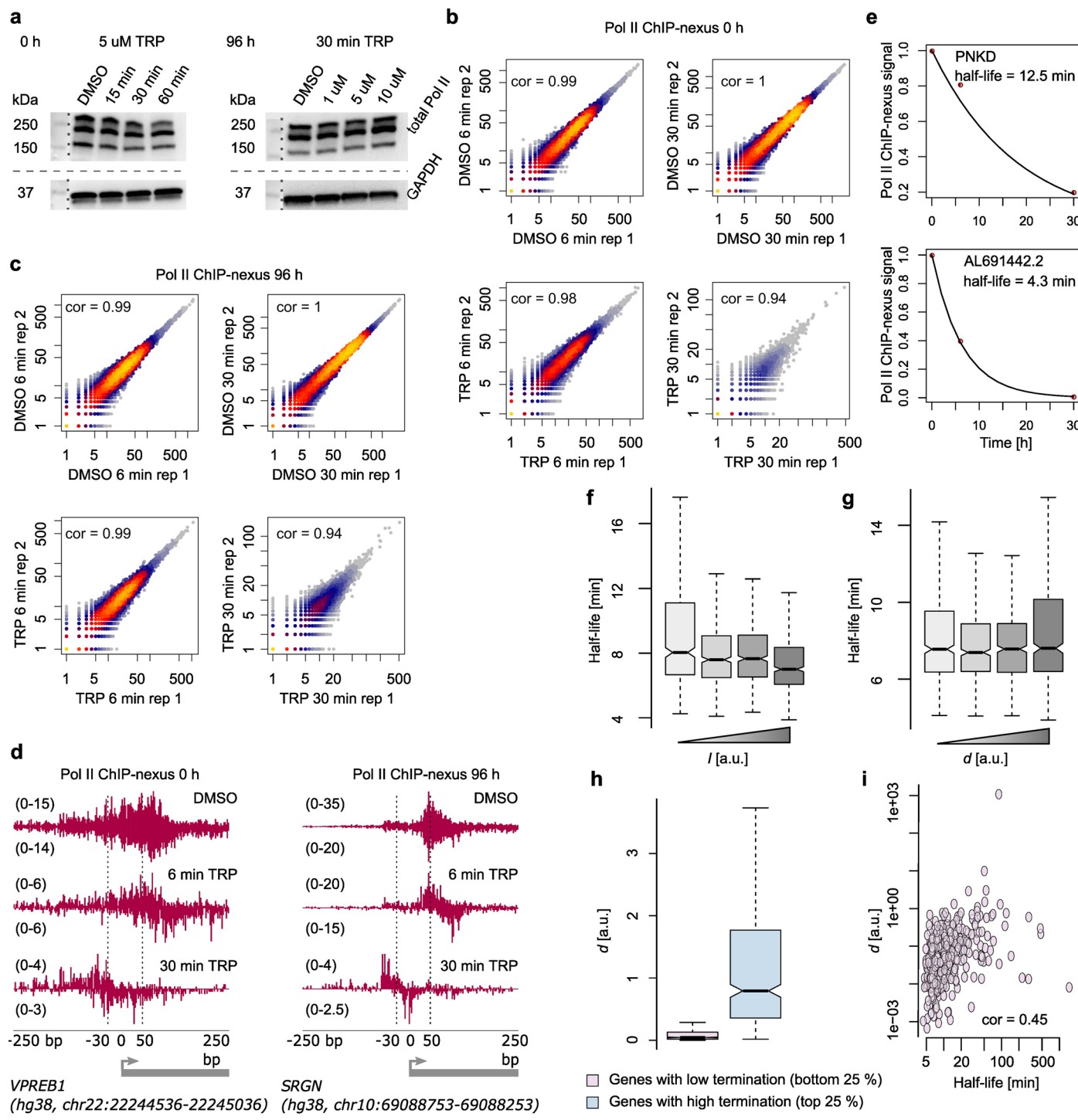

**Extended Data Fig. 4 | See next page for caption.**

**Extended Data Fig. 4 | Estimation of Pol II half-life in the promoter-proximal region. a**) Western Blotting analysis of the whole cell lysate obtained from DMSO/triptolide (TRP) treated cells at 0 h and 96 h of transdifferentiation. Treatment conditions are indicated in the legend. Total Pol II and GAPDH levels were detected. The experiment was repeated two times from independent biological replicates. Dashed lines represent membrane cut after the transfer. Dotted lines indicate where the molecular weight ladder is inserted, derived from the same membrane slice but under focused light. **b**) and **c**) Pearson correlation between two independent biological replicates of the ChIP-nexus data collected at 0 h (**b**) and 96 h (**c**) after transdifferentiation induction and treated with 5 µM of triptolide (TRP) or DMSO solvent control for 6 and 30 min. Heat color gradient as in Extended Data Fig. 1b. **d**) Representative gene examples of ChIP-nexus data after 6 and 30 min of DMSO or triptolide (TRP) treatment at 0 h and 96 h of transdifferentiation. ChIP-nexus data from two biological replicates were merged for illustration. **e**) Representative gene examples with different promoter-proximal Pol II half-lives at 96 h of transdifferentiation. Measurements of Pol II occupancy over the time course of triptolide (TRP) treatment were fitted to an exponential decay model (Methods). **f**) and **g**) Box plots showing promoter-proximal Pol II half-lives for four quantiles of estimated $l$ (**f**) and $d$ (**g**) values ranked from lowest to highest ($n = 1,814$). Estimates are based on two biological replicates. Data is shown for 96 h after transdifferentiation induction. Black bars represent medians, box limits are the first and third quartiles, and whiskers represent 1.5 times the interquartile range. Notches extend to 1.58 times the interquartile range divided by the square root of n (~95% confidence intervals of the median values). Outliers not shown. **h**) Box plots showing $d$ for groups of genes with low (bottom 25%, $n = 454$) and high (top 25%, $n = 454$) promoter-proximal Pol II termination fraction. Data is shown for 0 h after transdifferentiation induction. Box plot representations as in (**f**). **i**) Spearman correlation between $d$ and promoter-proximal Pol II half-life for genes with low (bottom 25%, $n = 454$) promoter-proximal Pol II termination fraction. Data is shown for 0 h after transdifferentiation induction.

# Reporting Summary

## Statistics

For all statistical analyses, confirm that the following items are present in the figure legend, table legend, main text, or Methods section.

| n/a | Confirmed | |
|---|---|---|
| ☐ | ☒ | The exact sample size (*n*) for each experimental group/condition, given as a discrete number and unit of measurement |
| ☐ | ☒ | A statement on whether measurements were taken from distinct samples or whether the same sample was measured repeatedly |
| ☐ | ☒ | The statistical test(s) used AND whether they are one- or two-sided *Only common tests should be described solely by name; describe more complex techniques in the Methods section.* |
| ☒ | ☐ | A description of all covariates tested |
| ☐ | ☒ | A description of any assumptions or corrections, such as tests of normality and adjustment for multiple comparisons |
| ☐ | ☒ | A full description of the statistical parameters including central tendency (e.g. means) or other basic estimates (e.g. regression coefficient) AND variation (e.g. standard deviation) or associated estimates of uncertainty (e.g. confidence intervals) |
| ☐ | ☒ | For null hypothesis testing, the test statistic (e.g. *F*, *t*, *r*) with confidence intervals, effect sizes, degrees of freedom and *P* value noted *Give P values as exact values whenever suitable.* |
| ☒ | ☐ | For Bayesian analysis, information on the choice of priors and Markov chain Monte Carlo settings |
| ☒ | ☐ | For hierarchical and complex designs, identification of the appropriate level for tests and full reporting of outcomes |
| ☐ | ☒ | Estimates of effect sizes (e.g. Cohen's *d*, Pearson's *r*), indicating how they were calculated |

*Our web collection on statistics for biologists contains articles on many of the points above.*

## Software and code

Policy information about availability of computer code

| | |
|---|---|
| Data collection | Sequencing data were collected using Illumina NextSeq 550. |
| Data analysis | Salmon v1.3.0; FASTQC v0.11.9; STAR v2.7.5a, Bowtie2 v2.3.4.1, Samtools v1.6, CutAdapt v2.3, R v4.2.0; DESeq2 v1.46.0, DAVID v2021q4; STRING db v11.5. Data analysis scripts: https://doi.org/10.5281/zenodo.14361017. |

For manuscripts utilizing custom algorithms or software that are central to the research but not yet described in published literature, software must be made available to editors and reviewers. We strongly encourage code deposition in a community repository (e.g. GitHub). See the Nature Portfolio guidelines for submitting code & software for further information.

## Data

Policy information about availability of data

All manuscripts must include a data availability statement. This statement should provide the following information, where applicable:
- Accession codes, unique identifiers, or web links for publicly available datasets
- A description of any restrictions on data availability
- For clinical datasets or third party data, please ensure that the statement adheres to our policy

Next-generation sequencing datasets generated in this study are available for download from GEO: GSE235181. Published TT-seq data used in this study is available for download from GEO: GSE131620.

# Research involving human participants, their data, or biological material

Policy information about studies with <u>human participants or human data</u>. See also policy information about <u>sex, gender (identity/presentation), and sexual orientation</u> and <u>race, ethnicity and racism</u>.

| | |
|---|---|
| Reporting on sex and gender | Not applicable |
| Reporting on race, ethnicity, or other socially relevant groupings | Not applicable |
| Population characteristics | Not applicable |
| Recruitment | Not applicable |
| Ethics oversight | Not applicable |

Note that full information on the approval of the study protocol must also be provided in the manuscript.

# Field-specific reporting

Please select the one below that is the best fit for your research. If you are not sure, read the appropriate sections before making your selection.

☒ Life sciences  ☐ Behavioural & social sciences  ☐ Ecological, evolutionary & environmental sciences

For a reference copy of the document with all sections, see <u>nature.com/documents/nr-reporting-summary-flat.pdf</u>

# Life sciences study design

All studies must disclose on these points even when the disclosure is negative.

| | |
|---|---|
| Sample size | All experiments were performed in two independent biological replicates. No statistical methods were used to pre-determine sample sizes, but our sample sizes are similar to those reported in previous publications by us (e.g., Choi et al., eLife 2021) and others (e.g., Shao et al., Nature Genetics 2017; Nojima et al., Cell, 2015). These sample sizes were chosen to generate data with sufficient depth and to robustly assess differences between time points. |
| Data exclusions | No data were excluded. Outliers were not drawn in the boxplots for clearer visualization. |
| Replication | All experiments involving genome-wide sequencing techniques were performed in two independent biological replicates. Western blotting and RT-qPCR experiments were independently repeated two times. All findings described in the manuscript were confirmed in all individual replicates. |
| Randomization | Cells were randomly seeded into the plates for the different time points of transdifferentiation for RT-qPCR, mNET-seq, ChIP- seq and ChIP-nexus experiments. Cells were randomly seeded into the plates for the different time points of DMSO and triptolide treatments for ChIP-nexus and Western blotting experiments. |
| Blinding | Blinding is not required. All samples were analysed using the same scripts and pipelines without any intervention by the investigator. Results were therefore directly related to the data and not influenced by any potential expectations of the researchers. |

# Reporting for specific materials, systems and methods

We require information from authors about some types of materials, experimental systems and methods used in many studies. Here, indicate whether each material, system or method listed is relevant to your study. If you are not sure if a list item applies to your research, read the appropriate section before selecting a response.

## Materials & experimental systems

| n/a | Involved in the study |
|---|---|
| ☐ | ☒ Antibodies |
| ☐ | ☒ Eukaryotic cell lines |
| ☒ | ☐ Palaeontology and archaeology |
| ☒ | ☐ Animals and other organisms |
| ☒ | ☐ Clinical data |
| ☒ | ☐ Dual use research of concern |
| ☒ | ☐ Plants |

## Methods

| n/a | Involved in the study |
|---|---|
| ☐ | ☒ ChIP-seq |
| ☒ | ☐ Flow cytometry |
| ☒ | ☐ MRI-based neuroimaging |

# Antibodies

| | |
|---|---|
| Antibodies used | mNET-seq: RNA Polymerase II antibody, monoclonal (MBL Life science, CMA601, MABI0601, C13B9) (30 µg were used per 2e8 BLaER1 cells); ChIP-seq: Cyclin T1 antibody, monoclonal (Cell Signaling, 81464, clone D1B6G) (12.5 µl were used per 50 µg of BLaER1 chromatin), CDK9 antibody, monoclonal (Abcam, ab239364, clone EPR22956-37) (7.9 µg were used per 50 µg of BLaER1 chromatin), Drosophila H2Av antibody (Active Motif, 61686) (0.5 µg was used per 122 ng Drosophila spike-ins per 50 µg of BLaER1 chromatin); ChIP-nexus: RNA Polymerase II NTD antibody, monoclonal (Cell Signaling, 14958, clone D8L4Y) (12 µL were used per 60 µg of BLaER1 chromatin), Drosophila H2Av antibody (Active Motif, 61686) (1 µg was used per 244 ng Drosophila spike-ins per 60 µg of BLaER1 chromatin); Western Blotting: RNA Polymerase II antibody, monoclonal (Santa-cruz, sc-55492, clone F-12) (used in 1:200 dilution), GAPDH antibody, monoclonal (Sigma-Aldrich, G8795, clone GAPDH-71.1) (used in 1:20,000 dilution), Goat Anti-Mouse IgG - H&L (HRP) antibody, polyclonal (Abcam, ab5870) (used in 1:3,000 dilution). |
| Validation | Validation of the antibodies was performed by the manufacturer. The information provided below is taken from the websites of the corresponding manufacturers. RNA Polymerase II antibody (MBL Life science, CMA601, MABI0601, C13B9; RRID:AB_2827956) was validated by ELISA. Cyclin T1 antibody (Cell Signaling, 81464; RRID:AB_2799973) was validated by western blotting, immunoprecipitation, ChIP and CUT&RUN techniques. CDK9 antibody (Abcam, ab239364; RRID:AB_3096172) was validated by ChIP-seq, ChIC/CUT&RUN, western blotting, immunoprecipitation, immunohistochemistry, ChIP-qPCR, immunocytochemistry/immunofluorescence and flow cytometry techniques. Drosophila H2Av antibody (Active Motif, 61686; RRID:AB_2737370) was validated by ChIP-qPCR and ChIP-seq techniques. RNA Polymerase II NTD antibody (Cell Signaling, 14958; RRID:AB_2687876) was validated by western blotting, ChIP-qPCR and ChIP-seq techniques. RNA Polymerase II antibody (Santa-cruz, sc-55492; RRID:AB_630203) was validated by western blotting, immunoprecipitation, immunohistochemistry, immunofluorescence and ELISA techniques. GAPDH antibody (Sigma-Aldrich, G8795; RRID:AB_1078991) was validated by western blotting, immunocytochemistry, immunofluorescence, indirect ELISA and microarray techniques. Goat Anti-Mouse IgG - H&L (HRP) antibody (Abcam, ab5870; RRID:AB_955389) was validated by dot blotting, electron microscopy, immunohistochemistry, western blotting, ELISA and immunocytochemistry/immunofluorescence techniques. |

# Eukaryotic cell lines

Policy information about cell lines and Sex and Gender in Research

| | |
|---|---|
| Cell line source(s) | The BLaER1 cell line was obtained from the laboratory of Thomas Graf (Rapino et al., Cell 2013). BLaER1 cells are a single subclone derived from C\EBPaER-GFP-transfected RCH-ACV B-cell leukemia cell line sorted for GFP expression (by the laboratory of Thomas Graf). |
| Authentication | Authentication was performed by FACS and transcriptome analysis by the laboratory of Thomas Graf (Rapino et al., 2013; Stick et al., 2020) and by our laboratory (Choi et al., 2021). STR profiling was performed by Millipore (see cat. # SCC165). Further information on BLaER1 authentication can be found in Cellosaurus, RRID:CVCL_VQ57. |
| Mycoplasma contamination | The BLaER1 cell line was regularly examined and tested negative for the mycoplasma contamination using Plasmo Test Mycoplasma Detection Kit (InvivoGen, rep-pt1). |
| Commonly misidentified lines (See ICLAC register) | No commonly misidentified cell lines were used. |

# Plants

| | |
|---|---|
| Seed stocks | Not applicable |
| Novel plant genotypes | Not applicable |
| Authentication | Not applicable |

# ChIP-seq

## Data deposition

☒ Confirm that both raw and final processed data have been deposited in a public database such as GEO.

☒ Confirm that you have deposited or provided access to graph files (e.g. BED files) for the called peaks.

| | |
|---|---|
| Data access links<br>*May remain private before publication.* | The sequencing data and processed files are deposited in the GEO database under accession code GSE235181. |
| Files in database submission | BLaER1_ChIPNexusseq_PolII_DMSO_30_0h_R1_S5_R2_001.minus.bw<br>BLaER1_ChIPNexusseq_PolII_DMSO_30_0h_R2_S6_R2_001.minus.bw |

```
BLaER1_ChIPNexusseq_PolII_DMSO_30_96h_R1_S1_R2_001.minus.bw
BLaER1_ChIPNexusseq_PolII_DMSO_30_96h_R2_S2_R2_001.minus.bw
BLaER1_ChIPNexusseq_PolII_DMSO_6_0h_R1_S1_R2_001.minus.bw
BLaER1_ChIPNexusseq_PolII_DMSO_6_0h_R2_S2_R2_001.minus.bw
BLaER1_ChIPNexusseq_PolII_DMSO_6_96h_R1_S1_R2_001.minus.bw
BLaER1_ChIPNexusseq_PolII_DMSO_6_96h_R2_S2_R2_001.minus.bw
BLaER1_ChIPNexusseq_PolII_TRP_30_0h_R1_S7_R2_001.minus.bw
BLaER1_ChIPNexusseq_PolII_TRP_30_0h_R2_S8_R2_001.minus.bw
BLaER1_ChIPNexusseq_PolII_TRP_30_96h_R1_S3_R2_001.minus.bw
BLaER1_ChIPNexusseq_PolII_TRP_30_96h_R2_S4_R2_001.minus.bw
BLaER1_ChIPNexusseq_PolII_TRP_6_0h_R1_S3_R2_001.minus.bw
BLaER1_ChIPNexusseq_PolII_TRP_6_0h_R2_S4_R2_001.minus.bw
BLaER1_ChIPNexusseq_PolII_TRP_6_96h_R1_S3_R2_001.minus.bw
BLaER1_ChIPNexusseq_PolII_TRP_6_96h_R2_S4_R2_001.minus.bw
BLaER1_ChIPNexusseq_PolII_DMSO_30_0h_R1_S5_R2_001.plus.bw
BLaER1_ChIPNexusseq_PolII_DMSO_30_0h_R2_S6_R2_001.plus.bw
BLaER1_ChIPNexusseq_PolII_DMSO_30_96h_R1_S1_R2_001.plus.bw
BLaER1_ChIPNexusseq_PolII_DMSO_30_96h_R2_S2_R2_001.plus.bw
BLaER1_ChIPNexusseq_PolII_DMSO_6_0h_R1_S1_R2_001.plus.bw
BLaER1_ChIPNexusseq_PolII_DMSO_6_0h_R2_S2_R2_001.plus.bw
BLaER1_ChIPNexusseq_PolII_DMSO_6_96h_R1_S1_R2_001.plus.bw
BLaER1_ChIPNexusseq_PolII_DMSO_6_96h_R2_S2_R2_001.plus.bw
BLaER1_ChIPNexusseq_PolII_TRP_30_0h_R1_S7_R2_001.plus.bw
BLaER1_ChIPNexusseq_PolII_TRP_30_0h_R2_S8_R2_001.plus.bw
BLaER1_ChIPNexusseq_PolII_TRP_30_96h_R1_S3_R2_001.plus.bw
BLaER1_ChIPNexusseq_PolII_TRP_30_96h_R2_S4_R2_001.plus.bw
BLaER1_ChIPNexusseq_PolII_TRP_6_0h_R1_S3_R2_001.plus.bw
BLaER1_ChIPNexusseq_PolII_TRP_6_0h_R2_S4_R2_001.plus.bw
BLaER1_ChIPNexusseq_PolII_TRP_6_96h_R1_S3_R2_001.plus.bw
BLaER1_ChIPNexusseq_PolII_TRP_6_96h_R2_S4_R2_001.plus.bw
BLaER1_ChIPseq_Input_CT1_CDK9_0h_Rep1.bw
BLaER1_ChIPseq_Input_CT1_CDK9_0h_Rep2.bw
BLaER1_ChIPseq_Input_CT1_CDK9_24h_Rep1.bw
BLaER1_ChIPseq_Input_CT1_CDK9_24h_Rep2.bw
BLaER1_ChIPseq_Input_CT1_CDK9_96h_Rep1.bw
BLaER1_ChIPseq_Input_CT1_CDK9_96h_Rep2.bw
BLaER1_ChIPseq_CDK9_0h_Rep1.bw
BLaER1_ChIPseq_CDK9_0h_Rep2.bw
BLaER1_ChIPseq_CDK9_24h_Rep1.bw
BLaER1_ChIPseq_CDK9_24h_Rep2.bw
BLaER1_ChIPseq_CDK9_96h_Rep1.bw
BLaER1_ChIPseq_CDK9_96h_Rep2.bw
BLaER1_ChIPseq_CT1_0h_Rep1.bw
BLaER1_ChIPseq_CT1_0h_Rep2.bw
BLaER1_ChIPseq_CT1_24h_Rep1.bw
BLaER1_ChIPseq_CT1_24h_Rep2.bw
BLaER1_ChIPseq_CT1_96h_Rep1.bw
BLaER1_ChIPseq_CT1_96h_Rep2.bw
```

Genome browser session
(e.g. UCSC)

Not applicable

## Methodology

**Replicates**

ChIP-seq and ChIP-nexus experiments were performed in two independent biological replicates. Detailed correlations between replicates are shown in the Extended Data section of this study.

**Sequencing depth**

The samples were sequenced using NextSeq550 Illumina platform to a depth of 30-40 million reads per sample for ChIP-seq experiments and to a depth of 80-100 million reads per sample for ChIP-nexus experiments.

**Antibodies**

ChIP-seq: Cyclin T1 antibody (Cell Signaling, 81464), CDK9 antibody (Abcam, ab239364), Drosophila H2Av antibody (Active Motif, 61686); ChIP-nexus: RNA Polymerase II NTD antibody (Cell Signaling, 14958), Drosophila H2Av antibody (Active Motif, 61686).

**Peak calling parameters**

Peak calling was performed by MACS2 with default parameters.

**Data quality**

The quality of the data was assessed by comparing the samples to available P-TEFb ChIP-seq data (Caizzi et al., Molecular Cell 2021) and Pol II ChIP-nexus data (Shao et al., Nature Genetics 2017).

**Software**

Paired-end reads of 75 bp length were collected for the samples. Reads were quality checked using FastQC. Reads were mapped to the human genome (GRCh38) using the Bowtie2 aligner. Other details about processing are described in the Methods section of this study.

