## [Peer Review File · Nature Structural & Molecular Biology]

Promoter-proximal RNA polymerase II termination regulates transcription during human cell type transition

Corresponding Author: Dr Michael Lidschreiber

Version 0:

Decision Letter:

12th Jan 2024

Dear Dr Lidschreiber,

Thank you again for submitting your manuscript "Promoter-proximal RNA polymerase II termination regulates transcription during human cell type transition". [I apologize for the delay in responding, which resulted from the difficulty in obtaining suitable referee reports. Nevertheless,] we now have comments (below) from the 3 reviewers who evaluated your paper. In light of those reports, we remain interested in your study and would like to see your response to the comments of the referees, in the form of a revised manuscript.

You will see that while the referees appreciate the potential interest of the novel findings and the overall quality of the data, all referees are concerned, to some extent, with the indirect nature of some of the data to support PPP vs termination effects. The referees #2 and #3 provide guidance to address this with additional data and analysis - please also revise the conclusions and discuss caveats in data interpretation accordingly.

Please be sure to address and respond to all concerns of the referees in full in a point-by-point response and highlight all changes in the revised manuscript text file. We are committed to providing a fair and constructive peer-review process. Do not hesitate to contact us if there are specific requests from the reviewers that you believe are technically impossible or unlikely to yield a meaningful outcome.

We appreciate the requested revisions are extensive. We thus expect to see your revised manuscript within 6 months. If you cannot send it within this time, please let us know. We will be happy to consider your revision as long as nothing similar has been accepted for publication at NSMB or published elsewhere. Should your manuscript be substantially delayed without notifying us in advance and your article is eventually published, the received date would be that of the revised, not the original, version.

Reporting Summary:

When submitting the revised version of your manuscript, please pay close attention to our [href="https://www.nature.com/nature-portfolio/editorial-policies/image-integrity">Digital Image Integrity Guidelines. and to the following points below:](https://www.nature.com/nature-portfolio/editorial-policies/image-integrity)

Please note that all key data shown in the main figures as cropped gels or blots must be presented in uncropped form, with molecular weight markers. These data can be aggregated into a single supplementary figure. While these data can be displayed in a relatively informal style, they must refer back to the relevant figures. These data should be submitted with the last revision, prior to acceptance, but you may want to start putting it together at this point.

SOURCE DATA: please provide, in tabular form, the data underlying the graphical representations used in figures. This is to further increase transparency in data reporting, as detailed in this editorial (<http://www.nature.com/nsmb/journal/v22/n10/full/nsmb.3110.html>). Spreadsheets can be submitted in excel format. Only one (1) file per figure is permitted; thus, for multi-paneled figures, the source data for each panel should be clearly labeled in the Excel file; alternately the data can be provided as multiple, clearly labeled sheets in an Excel file. When submitting files, the title field should indicate which figure the source data pertains to. We encourage our authors to provide source data at the revision stage, so that they are part of the peer-review process.

We require deposition of coordinates (and, in the case of crystal structures, structure factors) into the Protein Data Bank with the designation of immediate release upon publication (HPUB). Electron microscopy-derived density maps and coordinate data must be deposited in EMDB and released upon publication. Deposition and immediate release of NMR chemical shift assignments are highly encouraged. Deposition of deep sequencing and microarray data is mandatory, and the datasets must be released prior to or upon publication. To avoid delays in publication, dataset accession numbers must be supplied with the final accepted manuscript and appropriate release dates must be indicated at the galley proof stage. Please find the complete NRG policies on data availability at <http://www.nature.com/authors/policies/availability.html>.

Link Redacted

Sincerely,

Carolina Perdigoto, PhD
Chief Editor
Nature Structural & Molecular Biology
orcid.org/0000-0002-5783-7106

Referee expertise:

Referee #1: Pol II, RNA processing and transcription regulation

Referee #2: Pol II, transcription regulation and pausing

Referee #3: Pol II, RNA processing and transcription regulation

Reviewers' Comments:

Reviewer #1:

Remarks to the Author:

In essence these studies employ a trans-differentiation cell culture system in which pre-B leukemia cells are differentiated into macrophage-like cells by oestrogen over-expression. mNET-seq (made in this study) and TT-seq (previously made) analyses were performed on cells isolated during this trans-differentiation time course where mNET-seq provides profiles of transcriptionally active Pol II across protein coding genes while TT-seq measures de novo RNA synthesis as measured by 4thioU incorporation. In simple terms mNET-seq signals are higher near the promoter proximal regions due to Pol II "pausing" while TT-seq signals profile nascent transcriptional elongation in gene downstream regions. These analyses

identify genes that are downregulated during trans-differentiation as pre-B specific and those genes upregulated as iMAC specific. The detailed bioinformatics on these data sets has been performed to a high standard and allows a measure of transcriptional initiation (I) and promoter proximal pausing duration (d) as well as productive gene transcription. Previous analyses of transcriptional activation mechanisms have favoured a direct role for promoter proximal pause (PPP) release into productive transcription as a key activation mechanism. However more recently it is now well established that a substantial fraction PPP Pol IIs are directly turned over by termination mechanisms involving Integrator complex and/or recently described Restrictor complex mediated termination mechanisms (both strongly associated with lncRNA gene termination). As such transcriptional activation may more usually reflect new transcriptional initiation events which read through promoter proximal pause regions to achieve productive gene expression (especially as described by Larke et al Mol Cell. 2021)

The new data analyses as described in this study provide strong support for high levels of PPP associated termination and basically confirm the view that gene regulation is likely to be principally achieved by increased productive transcriptional initiation that reads through PPP regions. However, the predicted role for PPP mediated termination is only inferred indirectly by PPP Pol II stability (ChIP-seq) following transcriptional inhibition using triptolide. The likely side-effects of using this chemical treatment may lead to indirect effects (see below).

Further specific points:

1) Section entitled “pre-B genes show promoter-proximal transcription regulation”. There is arguably a lack of data to support this affirmation. The trans-differentiation time course reveals a decrease in TT-seq signal, used as a measure of elongation activity but a constant mNET-seq measured PPP signal. Also, decreased CDK9 and cyclin T1 (P-TEFb) CHIP-seq signals were observed.

These data only show that there is less gene initiation (less P-TEFb factors), that consequently leads to less elongation (less TT-seq signal). The fact that Pol II pausing remains constant throughout the time-course doesn't necessarily argue that promoter-proximal pausing regulates down-regulation of genes. Lower elongation levels can only be justified by the less Pol IIs initiating on the promoter. An alternative interpretation of the data could be that new Pol II needs to engage on promoters to create “pressure” on paused Pol II to transit into elongation. As soon as initiation is stopped Pol II pausing cannot be resolved.

2) Section entitled “Estimation of Pol II half-life in the promoter-proximal region” To make this estimation, triptolide is employed to block new transcriptional initiation. However, Triptolide inhibition could also significantly affect chromatin topology, associated TADs, as well as the affinity and likelihood of transcription factor binding. If engagement of new Pol IIs at promoters plays a direct role in the stability of promoter-proximity pausing, accurate measurement of real PPP stability (half-life) may be problematic. To understand if Pol II undergoes PPP or termination, simply blocking new initiation may be insufficient. This will only establish that PPP is stable when initiation is blocked. It won't determine PPP stability when new Pol IIs initiate transcription.

In summary I feel that this multi-omics approach provides some useful new data to better understand the regulation of gene expression through promoter proximal transcriptional events. However, the authors were unable to fully distinguish promoter proximal pausing from termination. The use of triptolide has limitations so that the combination of all the presented transcriptomics does not greatly advance our understanding of how PPP regulates gene expression at a mechanistic level.

Reviewer #2:

Remarks to the Author:

This manuscript by Lysakovskaia et al. investigates whether promoter-proximal termination by RNA polymerase II contributes to gene regulation. Termination of promoter-proximally paused RNA Pol II by complexes such as Integrator is well established, and it is known that depletion or degradation of Integrator subunits reduces this termination to affect gene activity, but whether termination is regulated under physiological conditions to control gene expression was not known. This work studies termination rates, as well as initiation and elongation rates, during cell differentiation, and shows that both gene activation and gene repression can involve altered termination frequencies. Thus, the authors conclude that promoter-proximal termination contributes to gene regulation.

This is a nicely executed study with a convincing set of conclusions. I have only two major concerns that should be addressed prior to publication.

1) I have concerns about the quality of the nascent RNA-seq data used in this study, and the way in which this data affects the authors terminology. The authors state that they can detect mNET-seq signal above background near the TSSs of only around half of the active genes in this study. They call the genes at which they can detect this signal ‘paused’. This implies that the other half of active genes do not undergo pausing, which I find extremely misleading.

I find it much more likely that the absence of nascent RNA signal near the other half of active gene promoters is due to the poor signal to noise of mNET-seq and/or misannotation of human TSSs. Certainly, prior analyses of nascent RNA signal using the more sensitive PRO-seq methodology has demonstrated the presence of paused RNA Pol II at not only all active genes, but even a subset of genes considered to be inactive!

To remedy this, I am not suggesting that the authors should re-do their nascent RNA experiments, but simply to change their language. Any reader of this manuscript will appreciate that the measurements made herein require significant signal for both TT-seq and promoter-proximal mNET-seq. So, filtering out the other active genes makes sense. I therefore suggest that the authors not define ‘paused’ and thus imply ‘not paused’ genes in Figure 1c, lines 101-102 or 117-118. Suggesting that so many active genes do not undergo pausing is inconsistent with the literature, including a number of important studies from the Cramer group itself. Instead, the authors should state that among the differentially expressed genes, 2,157 genes have sufficient TT-seq and mNET-seq signal at a dominant isoform to enable analysis and note that they will focus on this

set of genes.

2) All conclusions in this study are dependent on scripts that are not publicly available. This doesn't seem right to me. These scripts should be put on Github or similar and a Zenodo DOI should be provided so that others can make use of, and evaluate, these scripts.

Minor comments:

1) I am really perplexed by the spiky nature of the mNET seq data in the SLC9A9, VPRED3 and CD14 genes. What do the authors think this reflects? From the methods, it was not clear if the authors used UMIs to allow for removal of PCR duplicates. As shown recently by Andreas Mayer, such spikes in signal are often due to methodological artifacts, and NET-seq profiles can be substantially improved by both spike normalization and de-duplication. Can the authors clarify whether the data in this manuscript made use of either spike normalization or removal of PCR duplicates?

2) Extended Data Fig 4 d should include a scale bar to indicate by how many nucleotides the ChIP-nexus profiles shift upon TRP treatment. Can they include a line of 25nt in length, to provide readers with a sense for the resolution of these data?

Reviewer #3:

Remarks to the Author:

The manuscript by Lysakovskaia et al presents evidence for the importance of regulated premature termination of transcription for a subset of genes up and downregulated during differentiation of macrophages in a cell culture system. Although the evidence for premature termination is indirect and no details of its mechanism are reported, the paper is a valuable contribution to the field that illuminates the relative contributions of two different regulatory processes promoter proximal pausing and premature termination at a set of physiologically relevant genes.

Specific comments.

1. The definition of d as "apparent pause duration" is quite confusing especially to the non-specialist and should be changed to a more inclusive term such as perhaps \ggg
2. The data on PTEF-b occupancy as presented adds little to the paper (Extended Data Fig. 2c, d). It would be more effective to show metaplots of ChIP signals normalized to a spike in rather than heat plots.
3. Figure 3b presents critical data for distinct regulation of iMacI and iMacII gene groups which differ principally in the value of d at the $t=0$ time point (Fig. 3b lower right panel). Given the confusing definition of d , the authors should explain and illustrate more clearly this difference. It is not very evident from inspection of the TT-seq and mNETseq genome browser shots at $t=0$ h in Figure 3d and this apparent discrepancy between the results in Figs. 3b and 3d should be clarified.
4. The sentence in lines 265-268 is hard to follow. It would be helpful for the authors to explain explicitly the relation between "pol II half-life in the promoter-proximal region", and "premature termination fraction" as they are not independent of one another. It is rather counter-intuitive that the upregulated iMac II genes in Figure 5d, e show increased pol II turnover at 96hr yet decreased termination fraction since termination is a cause of turnover. The authors should clarify this point.
5. I did not find the data for downregulated genes that is referred to in the text (lines 277-282) as being in Figures 5c-e. As far as I could make out those figures refer only to the upregulated iMacI and iMac II gene sets.
6. The authors refer frequently to one review of premature termination (ref 16) but it would be more scholarly to acknowledge original papers that previously reported evidence of regulated premature termination at cellular and viral genes including: Kao, S.Y., Calman, A.F., Luciw, P.A., and Peterlin, B.M. (1987). Anti-termination of transcription within the long terminal repeat of HIV-1 by tat gene product. *Nature* 330, 489-493.
Brannan, K., Kim, H., Erickson, B., Glover-Cutter, K., Kim, S., Fong, N., Kiemele, L., Hansen, K., Davis, R., Lykke-Andersen, J., and Bentley, D.L. (2012). mRNA decapping factors and the exonuclease Xrn2 function in widespread premature termination of RNA polymerase II transcription. *Mol Cell* 46, 311-324. 10.1016/j.molcel.2012.03.006.
Wagschal, A., Rousset, E., Basavarajaiah, P., Contreras, X., Harwig, A., Laurent-Chabalier, S., Nakamura, M., Chen, X., Zhang, K., Meziane, O., et al. (2012). Microprocessor, Setx, Xrn2, and Rrp6 Co-operate to Induce Premature Termination of Transcription by RNAPII. *Cell* 150, 1147-1157. 10.1016/j.cell.2012.08.004.
Aoi, Y., Smith, E.R., Shah, A.P., Rendleman, E.J., Marshall, S.A., Woodfin, A.R., Chen, F.X., Shiekhattar, R., and Shilatifard, A. (2020). NELF Regulates a Promoter-Proximal Step Distinct from RNA Pol II Pause-Release. *Mol Cell* 78, 261-274 e265. 10.1016/j.molcel.2020.02.014.
Cugusi, S., Mitter, R., Kelly, G.P., Walker, J., Han, Z., Pisano, P., Wierer, M., Stewart, A., and Svejstrup, J.Q. (2022). Heat shock induces premature transcript termination and reconfigures the human transcriptome. *Mol Cell* 82, 1573-1588. 10.1016/j.molcel.2022.01.007.

Version 1:

Decision Letter:

10th Oct 2024

Dear Dr. Lidschreiber,

On behalf of my colleague, Dr. Carolina Perdigoto, thank you for your patience and for submitting your revised manuscript "Promoter-proximal RNA polymerase II termination regulates transcription during human cell type transition" (NSMB-A48486A). It has now been seen by the original referees and their comments are below. The reviewers find that the paper

has improved in revision, and therefore we'll be happy in principle to publish it in Nature Structural & Molecular Biology, pending minor revisions to satisfy Reviewer #1's final requests and to comply with our editorial and formatting guidelines.

We are now performing detailed checks on your paper and will send you a checklist detailing our editorial and formatting requirements in about 2-3 weeks (please note that this step takes a bit longer as it does require input from several production teams). Please do not upload the final materials and make any revisions until you receive this additional information from us.

To facilitate our work at this stage, it is important that we have a copy of the main text as a word file. If you could please send along a word version of this file as soon as possible, we would greatly appreciate it; please make sure to copy the NSMB account (nsmb@us.nature.com; cc'ed above).

Best regards,

George Inglis

George Inglis, PhD

Senior Editor

[Research Cross-Journal Editorial Team](https://www.nature.com/nsmb/research-cross-journal-editorial-team)
Nature Structural & Molecular Biology

Reviewer #1 (Remarks to the Author):

This revised ms from the Cramer lab addresses my previous concerns well.

I have however carefully reread the paper and remain somewhat confused by a few aspects of the detailed bioinformatic analyses which I raise below. At this stage I am hopeful that the authors can clarify my remaining issues with a view to enhancing understanding of the data presented especially for readers who are less computationally / statistically adept.

- 1) I am unsure why the ChIP-nexus data is used (Fig 4b, Fig 5b) as really mNET-seq analysis gives similar if not better Pol II positional information?
- 2) I don't see why the Fig 5b data shows (to my eye) identical profiles of TRP inhibition when we are shown (Fig 5e) that preB cell gave higher termination and iMacII lower termination after 96h TRP treatment. This should be commented on.
- 3) I finally note that the Zimmer et al Mol Cell paper really comes to very similar conclusions to this paper. In effect Zimmer et al show that TSS proximal termination is enhanced when genes are repressed and reduced when genes are activated. Of course, it is valuable to see this again in this different, leukemic B cell line trans-differentiation system.

Reviewer #2 (Remarks to the Author):

The authors have responded well to my suggestions and I now find this manuscript suitable for publication.

Reviewer #3 (Remarks to the Author):

All my criticisms were adequately addressed in the revised manuscript.

Version 2:

Decision Letter:

Dear Dr. Lidschreiber,

Thank you for your patience, we are now happy to accept your revised paper "Promoter-proximal RNA polymerase II termination regulates transcription during human cell type transition" for publication as a Article in Nature Structural & Molecular Biology.

After the grant of rights is completed, you will receive a link to your electronic proof via email with a request to make any

corrections within 48 hours. If, when you receive your proof, you cannot meet this deadline, please inform us at rjsproduction@springernature.com immediately.

Your paper will be published online soon after we receive proof corrections and will appear in print in the next available issue. You can find out your date of online publication by contacting the production team shortly after sending your proof corrections.

Happy New Year,

George

George Inglis, PhD

Senior Editor

[Research Cross-Journal Editorial Team](https://www.nature.com/nsmb/research-cross-journal-editorial-team)
Nature Structural & Molecular Biology

Click here if you would like to recommend Nature Structural & Molecular Biology to your librarian:

<http://www.nature.com/subscriptions/recommend.html#forms>

Reviewer #1:

Remarks to the Author:

In essence these studies employ a trans-differentiation cell culture system in which pre-B leukemia cells are differentiated into macrophage-like cells by oestrogen over-expression. mNET-seq (made in this study) and TT-seq (previously made) analyses were performed on cells isolated during this trans-differentiation time course where mNET-seq provides profiles of transcriptionally active Pol II across protein coding genes while TT-seq measures de novo RNA synthesis as measured by 4thioU incorporation. In simple terms mNET-seq signals are higher near the promoter proximal regions due to Pol II “pausing” while TT-seq signals profile nascent transcriptional elongation in gene downstream regions. These analyses identify genes that are downregulated during trans-differentiation as pre-B specific and those genes upregulated as iMAC specific. The detailed bioinformatics on these data sets has been performed to a high standard and allows a measure of transcriptional initiation (I) and promoter proximal pausing duration (d) as well as productive gene transcription.

Previous analyses of transcriptional activation mechanisms have favoured a direct role for promoter proximal pause (PPP) release into productive transcription as a key activation mechanism. However more recently it is now well established that a substantial fraction PPP Pol IIs are directly turned over by termination mechanisms involving Integrator complex and/or recently described Restrictor complex mediated termination mechanisms (both strongly associated with lncRNA gene termination). As such transcriptional activation may more usually reflect new transcriptional initiation events which read through promoter proximal pause regions to achieve productive gene expression (especially as described by Larke et al Mol Cell. 2021)

The new data analyses as described in this study provide strong support for high levels of PPP associated termination and basically confirm the view that gene regulation is likely to be principally achieved by increased productive transcriptional initiation that reads through PPP regions. However, the predicted role for PPP mediated termination is only inferred indirectly by PPP Pol II stability (ChIP-seq) following transcriptional inhibition using tryptolide. The likely side-effects of using this chemical treatment may lead to indirect effects (see below).

We would like to thank Reviewer #1 for the feedback and pointing out potential limitations. We have addressed the concerns as outlined in the following point-by-point responses.

Further specific points:

1) Section entitled “pre-B genes show promoter-proximal transcription regulation”. There is arguably a lack of data to support this affirmation. The trans-differentiation time course reveals a decrease in TT-seq signal, used as a measure of elongation activity but a constant mNET-seq measured PPP signal. Also, decreased CDK9 and cyclin T1 (P-TEFb) CHIP-seq signals were observed.

These data only show that there is less gene initiation (less P-TEFb factors), that consequently leads to less elongation (less TT-seq signal). The fact that Pol II pausing remains constant throughout the time-course doesn't necessarily argue that promoter-proximal pausing regulates down-regulation of genes. Lower elongation levels can only be justified by the less Pol IIs initiating on the promoter.

We agree with the reviewer that at this point we cannot conclude that promoter-proximal Pol II pausing regulates downregulation of pre-B genes. However, unchanged mNET-seq signal in the promoter-proximal region together with a decrease of productive RNA synthesis and a decrease of mNET-seq signal in the gene body can best be explained by two possible scenarios (See also Ehrensberger et al., Cell 2013, for a theoretical model behind this):

1) At 96 h, Pol II pauses longer in the promoter-proximal region of pre-B genes, thereby inhibiting new transcription initiation at the promoter (Shao and Zeitlinger, Nature Genetics 2017; Gressel et al., eLife 2017) resulting in the observed decrease of RNA synthesis. Under

this scenario, regulation of promoter-proximal Pol II pausing would be responsible for the downregulation of pre-B genes (via limiting/decreasing initiation at the promoter).

- 2) At 96 h, Pol II undergoes high levels of promoter-proximal termination for pre-B genes. An increase in promoter-proximal termination towards 96 h would allow for a decrease of productive RNA synthesis independent of a change in the initiation frequency at the promoter.*

We think that under both scenarios it is accurate to say that pre-B genes show “promoter-proximal regulation”. At this point we cannot conclude which of the two scenarios (or a combination of both) is true. We resolve this later, when we introduce additional measures and estimations of promoter-proximal Pol II half-life, promoter-proximal termination fraction and transcription initiation frequency at the promoter. We have extended the last sentence at the end of the respective paragraph to make this clearer.

As mentioned by the reviewer, a third scenario leading to downregulation of RNA synthesis would be regulation by changing only the initiation frequency at the promoter without changes in promoter-proximal regulation (i.e. promoter-proximal pausing and termination) similar to what was observed during erythropoiesis in mice (Larke et al., Molecular Cell 2021). This scenario is unlikely because, assuming there are no changes in promoter-proximal pause duration and termination, a decrease in initiation frequency at the promoter would lead to a corresponding decrease in promoter-proximal Pol II occupancy. However, we did not observe this in general. To further illustrate that downregulation of initiation frequency alone is unlikely, we simulated transcription of pre-B genes in silico and generated corresponding RNA synthesis (TT-seq) and transcriptionally engaged Pol II occupancy (mNET-seq) metagene profiles for 0 h and 96 h of transdifferentiation (Author Response Fig. 1a). The simulation was set up to reproduce the observed decrease in the RNA synthesis of pre-B genes by decreasing the initiation frequency at the promoter only, with no change in pause duration and promoter-proximal termination. The simulated metagene profiles show the expected decrease in Pol II promoter-proximal occupancy at 96 h, which is in contrast to what we observed in our experimental data (Author Response Fig. 1b).

Author Response Figure 1. Comparison of simulated RNA synthesis (TT-seq) and transcriptionally engaged Pol II occupancy (mNET-seq) metagene profiles for pre-B genes. The simulation tool is part of a manuscript that is currently in preparation and will be submitted elsewhere. Left: simulated profiles at 0 h and 96 h of transdifferentiation; the experimentally observed decrease in RNA synthesis of pre-B genes was recapitulated by decreasing only the initiation frequency at the promoter, leaving pause duration and promoter-proximal termination unchanged. Right: experimental profiles at 0 and 96 h of transdifferentiation (same as Fig. 2a in the manuscript).

An alternative interpretation of the data could be that new Pol II needs to engage on promoters to create “pressure” on paused Pol II to transit into elongation. As soon as initiation is stopped Pol II pausing cannot be resolved.

*This is an interesting alternative interpretation. Our ChIP-nexus data shows that while transcription initiation is inhibited, the promoter-proximal Pol II signal decreases over the time of inhibition (Fig. 4b, 5b and Extended Data Fig. 4d), implying that Pol II can still be released into productive elongation or terminate in the promoter-proximal region without “pressure” from initiating Pol II. This data is supported by another study that performed a similar experiment in *Drosophila* cells (Shao and Zeitlinger, Nature Genetics 2017). Additionally, the same study (Shao and Zeitlinger, Nature Genetics 2017) and two studies from our group (Gressel et al., eLife 2017; Gressel et al., Nature Communications 2019) investigated the relationship between transcription initiation and promoter-proximal pausing and experimentally showed that paused Pol II can limit transcription initiation rather*

than the other way around. Taken together, this implies that Pol II pausing can be resolved independently of Pol II initiation.

2) Section entitled “Estimation of Pol II half-life in the promoter-proximal region” To make this estimation, triptolide is employed to block new transcriptional initiation. However, Triptolide inhibition could also significantly affect chromatin topology, associated TADs, as well as the affinity and likelihood of transcription factor binding.

We appreciate the reviewer’s concern. We use short triptolide treatment times (6 and 30 min) with an optimized concentration that does not cause proteasomal Pol II degradation under these conditions (Extended Data Fig. 4a). It has been shown that triptolide treatment for 45 min does not cause significant changes in global chromatin organization, including compartments, TADs and loops in mouse embryonic stem cells (mESCs) (Hsieh et al., Molecular Cell 2020). Furthermore, a recent study showed that most chromatin microcompartments were not affected after 45 min and 4 h of triptolide treatment in mESCs (Goel et al., Nature Genetics 2023). Therefore, we believe that our cautious use of triptolide (short treatment times and optimized concentration) is well suited for promoter-proximal Pol II half-life measurements and does not affect chromatin organization.

We agree with the reviewer that triptolide can change the binding profiles of transcription initiation factors as it has been published previously (Shao and Zeitlinger, Nature Genetics 2017). However, this is expected since triptolide blocks the DNA translocase activity of TFIIH, resulting in stabilization of initiating Pol II at the promoter (Titov et al., Nature Chemical Biology 2011). To our knowledge, there is no data on how triptolide affects the binding of factors associated with the promoter-proximal region (in particular promoter-proximal pausing, pause release and termination factors).

If engagement of new Pol IIs at promoters plays a direct role in the stability of promoter-proximity pausing, accurate measurement of real PPP stability (half-life) may be problematic. To understand if Pol II undergoes PPP or termination, simply blocking new initiation may be insufficient. This will only establish that PPP is stable when initiation is blocked. It won’t determine PPP stability when new Pol IIs initiate transcription.

A recently published study (Zimmer et al., Molecular Cell 2021) used an alternative method to measure the half-lives of promoter-proximal Pol II without blocking initiation of Pol II and obtained a distribution of half-lives that is highly comparable to our measurements (compare Figure 6B in Zimmer et al., Molecular Cell 2021 to Fig. 4c in our manuscript). Therefore, we believe that our measurements provide reliable estimates of promoter-proximal Pol II stability.

In summary I feel that this multi-omics approach provides some useful new data to better understand the regulation of gene expression through promoter proximal transcriptional events. However, the authors were unable to fully distinguish promoter proximal pausing from termination. The use of triptolide has limitations so that the combination of all the presented transcriptomics does not greatly advance our understanding of how PPP regulates gene expression at a mechanistic level.

We hope that with our explanations we were able to convince the reviewer of the usefulness of our data and our approach as well as show that we are able to distinguish promoter-proximal termination from pausing.

Reviewer #2:

Remarks to the Author:

This manuscript by Lysakovskaia et al. investigates whether promoter-proximal termination by RNA polymerase II contributes to gene regulation. Termination of promoter-proximally paused RNA Pol II by complexes such as Integrator is well established, and it is known that depletion or degradation of Integrator subunits reduces this termination to affect gene activity, but whether termination is regulated under physiological conditions to control gene expression was not known. This work studies termination rates, as well as initiation and elongation rates, during cell differentiation, and shows that both gene activation and gene repression can involve altered termination frequencies. Thus, the authors conclude that promoter-proximal termination contributes to gene regulation.

This is a nicely executed study with a convincing set of conclusions. I have only two major concerns that should be addressed prior to publication.

We would like to thank Reviewer #2 for the positive feedback and suggestions for improvement. We have addressed the concerns regarding mNET-seq data quality and made all scripts available on Github.

1) I have concerns about the quality of the nascent RNA-seq data used in this study, and the way in which this data affects the authors terminology. The authors state that they can detect mNET-seq signal above background near the TSSs of only around half of the active genes in this study. They call the genes at which they can detect this signal 'paused'. This implies that the other half of active genes do not undergo pausing, which I find extremely misleading.

I find it much more likely that the absence of nascent RNA signal near the other half of active gene promoters is due to the poor signal to noise of mNET-seq and/or misannotation of human TSSs. Certainly, prior analyses of nascent RNA signal using the more sensitive PRO-seq methodology has demonstrated the presence of paused RNA Pol II at not only all active genes, but even a subset of genes considered to be inactive!

To remedy this, I am not suggesting that the authors should re-do their nascent RNA experiments, but simply to change their language. Any reader of this manuscript will appreciate that the measurements made herein require significant signal for both TT-seq and promoter-proximal mNET-seq. So, filtering out the other active genes makes sense. I therefore suggest that the authors not define 'paused' and thus imply 'not paused' genes in Figure 1c, lines 101-102 or 117-118. Suggesting that so many active genes do not undergo pausing is inconsistent with the literature, including a number of important studies from the Cramer group itself. Instead, the authors should state that among the differentially expressed genes, 2,157 genes have sufficient TT-seq and mNET-seq signal at a dominant isoform to enable analysis and note that they will focus on this set of genes.

We thank the reviewer for pointing this out. We agree that naming the selected genes as 'paused' is misleading and gives the wrong impression that other genes are not paused at all. Also, the way it was phrased in the main text could be misunderstood, as "mNET-seq signal enrichment above background" could give the impression that the remaining genes had no or only poor mNET-seq signal in the promoter-proximal region. However, this is not the case. In order to call our genes as 'paused' (or rather, to determine defined pause sites), the maximum mNET-seq signal in the promoter-proximal region (TSS to TSS + 250 bp) had to be at least five times greater than the median mNET-seq signal in this region (considering only non-zero values, Methods). This allowed us to select genes where a defined pause position could be clearly determined. Reducing this cutoff to e.g. 3-fold increases the number of selected genes by about 18%, but 5-fold was chosen as

described in previous publications (Gressel et al., eLife 2017; Gressel et al., Nature Communications 2019; Caizzi et al., Molecular Cell 2021). Overall, our mNET-seq data is of high quality and the vast majority of transcribed genes exhibit enriched promoter-proximal mNET-seq signal, consistent with previous literature describing promoter-proximal pausing as a ubiquitous process. We have carefully rephrased the respective paragraphs based on the reviewer's suggestion.

2) All conclusions in this study are dependent on scripts that are not publicly available. This doesn't seem right to me. These scripts should be put on Github or similar and a Zenodo DOI should be provided so that others can make use of, and evaluate, these scripts.

We agree with the reviewer and have made all scripts publicly available on Github (<https://github.com/cramerlab/Promoter-proximal-RNA-pol-II-termination-regulates-transcription-during-human-cell-transition-2024/>) and refer to the repository in the "Code availability" section of the revised manuscript.

Minor comments:

1) I am really perplexed by the spiky nature of the mNET seq data in the SLC9A9, VPRESB3 and CD14 genes. What do the authors think this reflects? From the methods, it was not clear if the authors used UMIs to allow for removal of PCR duplicates. As shown recently by Andreas Mayer, such spikes in signal are often due to methodological artifacts, and NET-seq profiles can be substantially improved by both spike normalization and de-duplication. Can the authors clarify whether the data in this manuscript made use of either spike normalization or removal of PCR duplicates?

We appreciate the reviewer's concern. As it was published before, PRO-seq can detect only paused and actively transcribing Pol II (Kwak et al., Science 2013), while mNET-seq can additionally detect backtracked Pol II complexes (Nojima et al., Cell 2015). This may explain, at least to some extent, the spikey nature of our mNET-seq data, which is generally no more spikey than other published (m)NET-seq data (see Author Response Fig. 2a). In support of this the newly developed LORAX-seq approach, which is able to detect backtracked Pol II complexes, was recently used to demonstrate the relationship between mNET-seq spikes and backtracked Pol II (Yang et al., BioRxiv 2023). Moreover, the NET-seq protocols used by the Mayer lab include analysis of nascent RNAs associated with the total chromatin fraction (Mayer et al., Cell 2015; Gajos et al., Nucleic Acids Research 2021), whereas the mNET-seq protocol we used includes an additional step of total Pol II immunoprecipitation from chromatin and further isolation of the associated nascent RNAs (Nojima et al., Cell 2015). Additionally, we used the detergent Empigen during the total Pol II immunoprecipitation step to wash out nascent RNA-associated splicing complexes that might give rise to unspecific mNET-seq signal as described (Schlackow et al., Molecular Cell 2017).

To test whether our mNET-seq data suffers from spike and/or duplication bias problems, we compared our mNET-seq data with nested NET-seq data from the Mayer lab (Gajos et al., Nucleic Acids Research 2021), which included UMI and spike normalization. The duplicate percentages of our mNET-seq data are in the range of 40-50%, which is comparable to mNET-seq experiments in previous studies by us and others (Author Response Fig. 2b). In comparison, the raw nested NET-seq data before UMI deduplication had duplicate percentages of around 96%, explaining the importance of UMI-based deduplication in their case. Pile-ups of NET-seq reads can also originate from products of mispriming during reverse transcription (RT) at the library preparation stage (Mayer et al., Cell 2015; Gajos et al., Nucleic Acids Research 2021). However, we could not detect enriched motifs in promoter-proximal mNET-seq peak regions that are similar to our RT primer sequence (Author Response Fig. 2c). Overall, we therefore believe that our mNET-seq data is of high quality

and well suited for the analyses and conclusions presented in our manuscript, even without UMI deduplication and spike normalization.

Finally, *SLC9A9* was not a very representative example for Fig. 3d, so we have changed it to *DAPP1*, also in response to Reviewer 3 point 3.

Author Response Figure 2.

- a)** Comparison of the spiky nature of NET-seq and mNET-seq data. IGV genome browser view of representative genes for the original mNET-seq (merged replicates, GSE60358, cyan, HeLa cells), mNET-seq with Empigen (GSE81662, cyan, HeLa cells), NET-seq (merged replicates, GSE162857, red, HEK293 cells), and nested NET-seq (merged replicates, GSE162857, red, HeLa cells) coverages compared to our mNET-seq data (merged replicates, 0 h, black). Published processed .bw coverage files were taken from GEO and lifted to hg38 where necessary.
- b)** Comparison of duplicate percentages of our mNET-seq data at 0 and 96 h, raw nested NET-seq data (GSE162857, Gajos et al., *Nucleic Acids Research* 2021) and other published mNET-seq datasets (Žumer et al., *Molecular Cell* 2021, Gressel et al., *eLife* 2017, Nojima et al., *Cell* 2015 / GSE60358). Duplicates are defined as the reads/read pairs (fragments) that map to the same genomic location with identical start and end positions. For biological replicates, the average percentage of duplicates is shown.
- c)** MEME (Timothy et al., *Nucleic Acids Research* 2015) was used for de novo motif discovery using the sequences of the promoter-proximal regions (TSS to TSS + 250 bp) of all analyzed genes with an annotated pause position ($n = 2157$, Fig. 1c). Only the first five motif hits from MEME are shown. Our RT primer sequence is shown below for comparison. In addition, we found that only 1.9% of the total genes analyzed had an exact match to the first 6-mer of our

RT primer in their promoter proximal regions, which was similar to the percentage we observed for background sequences (1.6% for all genes without annotated pause position).

- 2) Extended Data Fig 4 d should include a scale bar to indicate by how many nucleotides the CHIP-nexus profiles shift upon TRP treatment. Can they include a line of 25nt in length, to provide readers with a sense for the resolution of these data?

We agree with the reviewer that this would be helpful. We included a scale bar as suggested.

Reviewer #3:

Remarks to the Author:

The manuscript by Lysakovskaia et al presents evidence for the importance of regulated premature termination of transcription for a subset of genes up and downregulated during differentiation of macrophages in a cell culture system. Although the evidence for premature termination is indirect and no details of its mechanism are reported, the paper is a valuable contribution to the field that illuminates the relative contributions of two different regulatory processes promoter proximal pausing and premature termination at a set of physiologically relevant genes.

We would like to thank Reviewer #3 for the positive feedback and the suggestions for improvement. We have addressed the concerns as outlined in the following point-by-point responses.

Specific comments.

1. The definition of d as “apparent pause duration” is quite confusing especially to the non-specialist and should be changed to a more inclusive term such as perhaps >>>

We thank the reviewer for pointing this out. In our previous publications, we defined d as 'pause duration', which is the time required for Pol II to transcribe the pause region assuming no or little termination (Gressel et al., eLife 2017; Gressel et al., Nature Communications 2019). Therefore, for consistency with our previous publications, we decided to keep the term 'pause duration' but modify it to 'apparent pause duration' to account for the possibility of Pol II termination in the promoter-proximal region. We use the word 'apparent' because a $d = 5$ min could mean that on average one Pol II spends 5 min in the promoter-proximal region before being released into productive elongation, or it could mean, for example, that two Pol IIs each spend 2.5 min in the promoter-proximal region and one terminates. In other words, the pause duration d determined in our model reflects the total time that the promoter-proximal region is occupied by Pol II between two consecutive productive initiation events (i.e., that successfully lead to productive elongation). This does not necessarily mean that one polymerase pauses for the entire estimated time, but could also mean that a subpopulation of polymerases at the pause site terminates early. In general, however, we agree with the reviewer's concern about non-specialist readers and would consider an alternative term if suggested. We have also added a sentence to clarify the meaning of d in the first section of the results.

2. The data on PTEF-b occupancy as presented adds little to the paper (Extended Data Fig. 2c, d). It would be more effective to show metaplots of ChIP signals normalized to a spike in rather than heat plots.

We thank the reviewer for this comment. We have changed the plots accordingly.

3. Figure 3b presents critical data for distinct regulation of iMacI and iMacII gene groups which differ principally in the value of d at the $t=0$ time point (Fig. 3b lower right panel). Given the confusing definition of d , the authors should explain and illustrate more clearly this difference. It is not very evident from inspection of the TT-seq and mNETseq genome browser shots at $t=0$ h in Figure 3d and this apparent discrepancy between the results in Figs. 3b and 3d should be clarified.

We thank the reviewer for this observation. Because the calculation of d includes both the productive initiation frequency from TT-seq and the transcriptionally engaged Pol II occupancy from mNET-seq, it is generally difficult to clearly illustrate differences in d when showing individual TT-seq and mNET-seq coverage profiles of example genes. In the revised manuscript we have addressed the

reviewer's concern by adding additional plots to the extended data figures that show the overlay of iMac I and II gene example profiles at 0 h and 96 h using the same y-scale and also zooming into the promoter-proximal region (plotted from TSS to TSS + 200) for the mNET-seq profile (Extended data Fig. 3b). For iMac II genes, we chose DAPP1 instead of SLC9A9 as example gene to better reflect the differences in d and also to address Reviewer 2's minor point 1. In addition, we report the values of d for these two example genes at 0 h and 96 h in the respective figure legend.

The fact that d values are generally higher for iMac II genes compared to iMac I genes at $t=0$ h means that Pol II either pauses longer and/or terminates more frequently in the promoter-proximal region of iMac II genes at this time point. At this point in the manuscript, we cannot conclude which of the two scenarios (or a combination of both) is true. We resolve this later, when we introduce additional measures and estimations of promoter-proximal Pol II half-life and promoter-proximal termination fraction. We have added a sentence to clarify the meaning of d in the first section of the results. In addition, we have added two more sentences to the results section that describe and clarify the observed difference in d at $t=0$ h between iMac I and II genes based on Fig. 3b-d and the new Extended data Fig. 3b.

4. The sentence in lines 265-268 is hard to follow. It would be helpful for the authors to explain explicitly the relation between "pol II half-life in the promoter-proximal region", and "premature termination fraction" as they are not independent of one another. It is rather counter-intuitive that the upregulated iMac II genes in Figure 5d, e show increased pol II turnover at 96hr yet decreased termination fraction since termination is a cause of turnover. The authors should clarify this point.

The reviewer correctly points out that termination is also a cause of turnover. The total turnover rate shown in Fig. 5d is the sum of the rates of release of promoter-proximal Pol II into productive elongation and termination. Productive initiation frequency (I) provides an estimate of the number of promoter-proximal Pol II that is released into productive elongation in a given period of time. This quantity, combined with the number of Pol II terminated in the promoter-proximal region in the given period of time, gives the total Pol II turnover rate (r) during that time, which we obtain from the estimated half-lives of promoter-proximal Pol II (using the Pol II ChIP-nexus experiments during a time course of initiation inhibition). The difference between r and I gives us an estimate of the number of promoter-proximal Pol II undergoing termination in the given period of time. In Fig. 5e, we represent this difference as a fraction of the total turnover rate and call it the termination fraction ($1 - I/r$). The termination fraction provides information about the proportion of total initiated Pol II that was terminated in the promoter-proximal region at each time point. Now, one can imagine a situation where the total number of initiating Pol II has increased while the fraction of those going into termination has decreased, resulting in a greater increase in productive initiation frequency (release of Pol II into productive elongation). This explains the observed increase in the total Pol II turnover rate despite a decrease in the termination fraction. Moreover, this is in contrast to a scenario where only the total number of initiating Pol II increases without a change in the termination fraction. In our study, we show that the iMac II and iMac I genes are representative of these two scenarios (Fig. 5d, e).

5. I did not find the data for downregulated genes that is referred to in the text (lines 277-282) as being in Figures 5c-e. As far as I could make out those figures refer only to the upregulated iMacI and iMac II gene sets.

The downregulated pre-B genes are shown in green in Figure 5. We clarified this in the legend and made the labels more visible.

6. The authors refer frequently to one review of premature termination (ref 16) but it would be more scholarly to acknowledge original papers that previously reported evidence of regulated premature termination at cellular and viral genes including:

Kao, S.Y., Calman, A.F., Luciw, P.A., and Peterlin, B.M. (1987). Anti-termination of transcription within the long terminal repeat of HIV-1 by tat gene product. *Nature* 330, 489-493.

Brannan, K., Kim, H., Erickson, B., Glover-Cutter, K., Kim, S., Fong, N., Kiemele, L., Hansen, K., Davis, R., Lykke-Andersen, J., and Bentley, D.L. (2012). mRNA decapping factors and the exonuclease Xrn2 function in widespread premature termination of RNA polymerase II transcription. *Mol Cell* 46, 311-324. [10.1016/j.molcel.2012.03.006](https://doi.org/10.1016/j.molcel.2012.03.006).

Wagschal, A., Rousset, E., Basavarajaiah, P., Contreras, X., Harwig, A., Laurent-Chabalier, S., Nakamura, M., Chen, X., Zhang, K., Meziane, O., et al. (2012). Microprocessor, Setx, Xrn2, and Rrp6 Co-operate to Induce Premature Termination of Transcription by RNAPII. *Cell* 150, 1147-1157. [10.1016/j.cell.2012.08.004](https://doi.org/10.1016/j.cell.2012.08.004).

Aoi, Y., Smith, E.R., Shah, A.P., Rendleman, E.J., Marshall, S.A., Woodfin, A.R., Chen, F.X., Shiekhattar, R., and Shilatifard, A. (2020). NELF Regulates a Promoter-Proximal Step Distinct from RNA Pol II Pause-Release. *Mol Cell* 78, 261-274 e265. [10.1016/j.molcel.2020.02.014](https://doi.org/10.1016/j.molcel.2020.02.014).

Cugusi, S., Mitter, R., Kelly, G.P., Walker, J., Han, Z., Pisano, P., Wierer, M., Stewart, A., and Svejstrup, J.Q. (2022). Heat shock induces premature transcript termination and reconfigures the human transcriptome. *Mol Cell* 82, 1573-1588. [10.1016/j.molcel.2022.01.007](https://doi.org/10.1016/j.molcel.2022.01.007).

We thank the reviewer for the reference suggestions. We have added all references from the list that show evidence of promoter-proximal Pol II termination.

Reviewer #1 (Remarks to the Author):

This revised ms from the Cramer lab addresses my previous concerns well.

I have however carefully reread the paper and remain somewhat confused by a few aspects of the detailed bioinformatic analyses which I raise below. At this stage I am hopeful that the authors can clarify my remaining issues with a view to enhancing understanding of the data presented especially for readers who are less computationally / statistically adept.

1) I am unsure why the ChIP-nexus data is used (Fig 4b, Fig 5b) as really mNET-seq analysis gives similar if not better Pol II positional information?

We used ChIP-nexus data for two reasons. First, this had been successfully used by others to estimate the promoter-proximal half-life of Pol II (Shao and Zeitlinger, Nature Genetics 2017), and second, unlike mNET-seq, ChIP-nexus provides information on Pol II occupancy not only downstream of the TSS, but also upstream of the TSS where the preinitiation complex assembles at the promoter. This allowed us to observe the shift of the Pol II peak from the promoter-proximal region to upstream of the TSS, further validating the functional inhibition of initiation by triptolide. However, the reviewer is correct that we could have used mNET-seq to obtain similar, if not better, Pol II positional information downstream of the TSS. A recent study, published after our ChIP-nexus experiments and analysis were completed, used mNET-seq and triptolide treatment to estimate promoter-proximal Pol II half-lives in mouse embryonic stem cells (Wang et al., Nature 2023). Importantly, the distribution of promoter-proximal Pol II half-lives they obtained in steady-state mESC cells was very similar to that observed in our measurements in human BLaER1 cells (compare Fig. 3h in Wang et al. with 4c in our study). Therefore, we believe that our results and conclusions are unaffected by the choice of ChIP-nexus over mNET-seq.

2) I don't see why the Fig 5b data shows (to my eye) identical profiles of TRP inhibition when we are shown (Fig 5e) that preB cell gave higher termination and iMacII lower termination after 96h TRP treatment. This should be commented on.

The reviewer correctly notes the visual similarity between the triptolide inhibition Pol II ChIP-nexus profiles that we have shown in Figure 5b. This similarity is also reflected in the half-lives shown in Figure 5c, which were estimated from these data. The similarity arises because 1. most of the genes we analyzed have similar half-lives (e.g. pre-B and iMac II after 96h TRP treatment, as mentioned by the reviewer), and 2. the axis limits of the plots in Figure 5b are not the same. The reason why genes with similar TRP inhibition Pol II ChIP-nexus profiles, and thus similar half-lives, can show differences in promoter-proximal termination is simply that the calculation of the Pol II termination fraction includes not only the half-life but also the promoter-proximal occupancy of Pol II (to estimate the total Pol II turnover r , Fig. 5a, Methods) as well as the productive initiation frequency I (Fig. 5a, Methods), which is based on the TT-seq signal. In other words, similar half-lives do not imply similar termination fractions, as the latter also depend on promoter-proximal Pol II occupancy and productive initiation frequency, both of which are different for the gene groups at 0 and 96 h (Fig. 5b, Fig. 2, Fig. 3). For example, in the case of downregulated pre-B genes, TT-seq signal, i.e. productive initiation frequency I , decreases (Fig. 2c, left panel), while promoter-proximal Pol II occupancy (Fig. 2c, middle panel) and Pol II half-life (Fig. 5c) remain unchanged from 0 to 96h. This can only be explained by increased termination (Fig. 5e) since if termination were unchanged for these genes, this would have led to an increase in Pol II half-life, which we did not observe. See also reviewer 1, point 1 in the previous response letter for a detailed explanation (including simulation) of why the decrease in TT-seq signal cannot be explained solely by a decrease in initiation at the promoter. How we derive the termination fraction is described in the main results section "Estimation of the Pol II termination fraction in the promoter-proximal region" (also highlighted in Figure 5a), using less computationally adept language as the reviewer requested, and in the Methods. We have made some more minor changes to further improve clarity.

3) I finally note that the Zimmer et al Mol Cell paper really comes to very similar conclusions to this paper. In effect Zimmer et al show that TSS proximal termination is enhanced when genes are repressed and reduced when genes are activated. Of course, it is valuable to see this again in this different, leukemic B cell line trans-differentiation system.

We appreciate the reviewer's comment in this regard. The paper by Zimmer et al, 2021 Molecular Cell, did indeed examine promoter-proximal Pol II termination under hyperosmotic stress in human cells. Based on their observations, they suggested that increased promoter-proximal Pol II termination is responsible for gene downregulation. The remaining conclusions of this paper are based primarily on data from Drosophila. We agree with the reviewer that it is valuable that we reach similar conclusions regarding gene downregulation using a different approach in a different human system. Furthermore, in our human leukemic B-cell transdifferentiation system, we also show a role for promoter-proximal Pol II termination in gene upregulation, with clear differences between gene groups.